# Variability and trends in the PV-gradient dynamical tropopause

Katharina Turhal[1,2], Felix Plöger[1,2], Jan Clemens[1,2], Thomas Birner[3,4], Franziska Weyland[5], Paul Konopka[1], and Peter Hoor[5]

[1]Institute of Climate and Energy Systems, Stratosphere (ICE-4), Forschungszentrum Jülich, Jülich, Germany.
[2]Institute for Atmospheric and Environmental Research, University of Wuppertal, Wuppertal, Germany.
[3]Meteorological Institute, Ludwig-Maximilians-Universität München, Munich, Germany.
[4]Institute of Atmospheric Physics, Deutsches Zentrum für Luft- und Raumfahrt (German Aerospace Center), Oberpfaffenhofen, Germany.
[5]Institute for Atmospheric Physics, Johannes Gutenberg University Mainz, Mainz, Germany.

**Correspondence:** Katharina Turhal (k.turhal@fz-juelich.de)

**Abstract.** The dynamical tropopause acts as a transport barrier between the tropical upper troposphere and extratropical low-ermost stratosphere and is characterized by steep gradients in potential vorticity (PV) along an isentropic surface. Hence, the latitudinal separation between the dynamical tropopause in the northern and southern hemispheres can be used as a metric of upper tropospheric width for assessing climate change impacts. Here, we obtain the PV gradient-based dynamical tropopause (PVG tropopause) from four meteorological satellite-era reanalyses (ERA5, ERA-Interim, JRA-55, MERRA-2) and investigate its climatology, variability and long-term trends ranging from 1980 to 2017. Our results show a distinct seasonal cycle with larger PV values and a poleward movement of the PVG tropopause in summer. The climatological tropopause PV values are substantially different between different reanalyses, but the tropopause latitude is similar. Significant interannual variability in the PVG tropopause latitude is related to El Niño Southern Oscillation (ENSO) and weaker variability also to the Quasi-Biennial Oscillation (QBO), and is consistently represented in reanalyses. In particular, El Niño correlates with equatorward shifts of the PVG tropopause, hence a decrease of upper tropospheric width. Long-term trends in the PVG tropopause over the period 1980–2017 exhibit a distinct vertical structure with poleward shifts below $340\,\mathrm{K}$ potential temperature, equatorward shifts between $340\,\mathrm{K}$ to $370\,\mathrm{K}$ and poleward shifts between $370\,\mathrm{K}$ to $380\,\mathrm{K}$. This consistent widening at lower levels and narrowing in the upper troposphere exhibits also considerable zonal variability with strongest upper tropospheric narrowing over the East Pacific.

## 1 Introduction

The transition between the convective, well-mixed troposphere and more stably layered stratosphere occurs at the *tropopause*, which is characterized by a sharp temperature inversion, giving rise to the common definition of the *thermal tropopause* based on a lapse rate criterion (World Meteorological Organization, 1957). The tropopause height increases from around $8\,\mathrm{km}$ at the

poles until around $17\,\text{km}$ near the equator. In the tropics to mid-latitudes, the tropopause altitude is strongly influenced by the tropospheric *Hadley circulation*, which forms due to differential solar heating and baroclinic instability in the mid-latitudes. In the innertropical convergence zone (ITCZ), large-scale convective updraft of humid airmasses raises the tropopause to altitudes between $12\,\text{km}$ to $18.5\,\text{km}$ (Fujiwara et al., 2022, Chap. 8). The poleward flow aloft is deflected by the Coriolis force, creating bands of westerly winds, the *subtropical jet streams (STJ)*, which are typically located around $20°$ to $40°$ latitude in each hemisphere and $12\,\text{km}$ height (Manney and Hegglin, 2018). The subtropical jets act as waveguides for Rossby waves, which travel eastwards with the stream. Upon breaking, these waves induce large-scale stirring and small-scale turbulence, transferring momentum and heat, which reinforces the Hadley circulation—an effect known as "eddy pump" (Staten et al., 2018). Large-scale downwelling of the Hadley circulation in the subtropics causes the tropopause to drop sharply in altitude and occasionally become discontinuous, referred to as the *subtropical tropopause break*, which coincides with the STJ.

In the mid-latitudes, warm subtropical airmasses encounter cold polar air, creating a turbulent baroclinic front. Eddies in this baroclinity zone offset the angular momentum and heat budgets of the Hadley circulation and are therefore closely correlated to the extent and strength of the Hadley cells (e. g., Korty and Schneider, 2008; Davis and Birner, 2019). Furthermore, the baroclinic fronts give rise to the *eddy-driven jets (EDJ)* of strong westerly winds in each hemisphere, which are usually located poleward and at altitudes slightly below the subtropical jets, between $40°$ to $70°$ latitude and around $10\,\text{km}$ height (Manney and Hegglin, 2018). Occasionally, the subtropical and eddy-driven jets are collocated and exhibit only one wind maximum (e. g., Lee and Kim, 2003; Archer and Caldeira, 2008).

In potential temperature coordinates, the tropopause mostly ranges between the isentropes of 300 K near the poles and 380 K in the tropics. The region between the extratropical tropopause and the 380 K surface is referred to as the *lowermost stratosphere*, which is an important region with regards to stratosphere-troposphere exchange (STE), as isentropes cross the tropopause there, enabling quasi-adiabatic horizontal transport between the tropical troposphere and extratropical stratosphere (Holton et al., 1995).

Another common definition of the tropopause, the so-called *dynamical tropopause*, is based on the *potential vorticity (PV)*, an analogue of angular momentum in air flow introduced by Rossby (1940) and Ertel (1942). PV is measured in *Potential Vorticity Units (PVU)* with $1\,\text{PVU} = 10^{-6}\,\text{m}^2\text{s}^{-1}\text{K}\,\text{kg}^{-1}$. An invertibility principle holds which allows inferring the flow velocity field from the PV distribution. PV is therefore closely linked to atmospheric dynamics (Hoskins et al., 1985), which makes it particularly valuable for transport studies.

At the tropopause, there is a transition from lower PV values in the troposphere to higher PV values in the stratosphere due to the increase of static stability (Hoskins, 1991). As PV is conserved in large-scale adiabatic and frictionless flow, these steep PV gradients in the subtropical tropopause represent a transport barrier to isentropic, quasi-adiabatic exchange between the upper troposphere and the lowermost stratosphere (Holton et al., 1995) as well as to eddy mixing across the subtropical jet stream (e. g., Haynes and Shuckburgh, 2000).

A PV-based *dynamical tropopause* was introduced by Reed (1955) and further developed by Danielsen (1964) and Shapiro (1980) in order to distinguish tropospheric and stratospheric air during tropopause folds. In many following transport studies, PV isosurfaces between $1.5\,\text{PVU}$ to $4\,\text{PVU}$ have been employed as the dynamical tropopause, as this PV range often cor-

responds to the location of the strongest PV gradient (e. g., Hoerling et al., 1991; Holton et al., 1995). However, Kunz et al. (2011) showed that PV at the tropopause is subject to strong variations, questioning the applicability of single PV isosurfaces as a tropopause definition. Therefore, defining the tropopause via the PV gradient and subtropical jet strength is a promising definition and potentially useful approach for studies of upper troposphere-lower stratosphere (UTLS) transport.

The isentropic gradient of PV as a characteristic to define stratospheric transport barriers has first been used by Nash et al. (1996) to describe the edge of the stratospheric polar vortex. Similarly, for the upper-level anticyclonic Asian monsoon circulation, isentropic PV gradients have proven successful to describe differences between atmospheric regions of different chemical composition and to define the core of the anticyclonic monsoon circulation (Ploeger et al., 2015). This concept has also been employed by Kunz et al. (2011, 2015) to define the extratropical tropopause from the maximum subtropical PV gradient along

an isentropic surface. This dynamical PV gradient-based tropopause, hereafter abbreviated as *PVG tropopause*, can be calculated on each upper-tropospheric isentropic level and in each hemisphere. Consequently, the latitudinal separation between the PVG tropopause in the northern and southern hemispheres can be used as a level-sensitive metric for the width of the upper troposphere; it offers the opportunity to investigate changes in tropospheric width at different levels, which may be a useful metric for assessing changes in tropical circulation and the width of the tropics.

Various metrics have been employed for the width of the tropical belt, considering upper-tropospheric variables such as the tropopause break or subtropical jet latitude, as well as lower-tropospheric variables e. g., the Hadley cell edge, precipitation or surface wind minima in the subtropics (Staten et al., 2018). In most of the tropical width metrics, an expansion of the tropics within the last 30 to 40 years (since the beginning of global satellite observations around 1979) has been detected in numerous studies using observations and reanalysis data (e. g., Seidel and Randel, 2007; Hu and Fu, 2007; Fu and Lin, 2011) as well as

model simulations (e. g., Lu et al., 2007; Frierson et al., 2007; Johanson and Fu, 2009; Hu et al., 2013; Solomon et al., 2016). Such an expansion of tropical width potentially has severe consequences for the ecology and population of broad regions in the subtropics, e. g., by shifting precipitation patterns (Si et al., 2009; Brönnimann et al., 2015) and cyclone tracks (Studholme and Gulev, 2018), intensifying droughts and the expansion of drylands (Post et al., 2014; Scheff and Frierson, 2012; Feng and Fu, 2013).

Initial estimates of tropical widening ranged from $0.25°$ to $3°$ latitude per decade (Seidel et al., 2008) with considerable variations between the different metrics. The larger trends have later been attributed to inhomogeneities in older reanalyses (Davis and Davis, 2018) and internal variability (Grise et al., 2018). Using recent reanalyses and taking into account internal variabilities, different metrics yielded an expansion of the tropical belt with a rate of $0.25°$ to $0.5°$ latitude per decade, which is consistent with model simulations (Staten et al., 2020). However, the expansion observed in near-surface metrics is often not

consistent with upper-tropospheric metrics such as the tropopause break or subtropical jet latitudes, which also disagree among each other in magnitude and even sign (Waugh et al., 2018).

The possible causes and mechanisms driving the expansion observed in different tropical width metrics are still under discussion, including the increase of greenhouse gas concentrations, depletion of stratospheric ozone, as well as both natural and anthropogenic aerosol (Staten et al., 2018, 2020). The coupled natural variability of sea surface temperatures and the

atmosphere, such as El Niño Southern Oscillation (ENSO) and the closely linked Pacific Decadal Oscillation (PDO), also

affects tropical width. During El Niño, the Hadley cells contract and the inter-hemispheric distance between the subtropical jets is reduced (Lu et al., 2008; Chen et al., 2008; Staten et al., 2018). The negative PDO phase during recent decades probably lead to a widening of the Hadley cells (Allen and Kovilakam, 2017; Grise et al., 2019; Staten et al., 2018). The role of internal variability is still under discussion; some studies indicate that the impact of natural variability even exceeds that of long-term changes, and that the currently available 40 years of reanalysis data are too short to discern forced expansion from natural variability (Grise et al., 2018; Staten et al., 2020).

Tropical width variability implies changes in the structure of the upper troposphere and lower stratosphere with further consequences for trace gas transport and composition which, in turn, may cause feedbacks on circulation and climate via radiative effects. Regarding tropopause variability, from radiosonde observations and reanalyses a general rise in the tropical tropopause has been determined since around 1980 (Seidel et al., 2001; Santer et al., 2003; Xian and Homeyer, 2019; Zou and Hoffmann, 2023). Notably, Zou and Hoffmann (2023) determined a general rise and cooling of the tropical tropopause in ERA5 between 1980 and 2021, but also a decline in this rising trend since the late 1990s. Tropical width trends derived from the tropical tropopause indicate an overall tropical narrowing of (-0.16 $\pm$ 0.11)° per decade between 1980 and 2021, with a widening period between 1980–2005 and narrowing after 2006 (Zou and Hoffmann, 2023).

Seidel and Randel (2007) introduced a tropical width metric based on the frequency of high tropopause altitudes, indicating a rise and expansion of the tropical tropopause in this and several following studies (Lu et al., 2009; Lucas et al., 2012; Davis and Rosenlof, 2012) which has been found to be sensitive to height thresholds and yielding ambiguous results (Birner, 2010). Davis and Rosenlof (2012) furthermore introduced a method which determines the latitude of the subtropical tropopause break region, using the maximum gradient in tropopause height in an area-weighted mean. Other studies relying on the tropopause break have been carried out by Davis and Birner (2013, 2017) and Martin et al. (2020). Martin et al. (2020) found substantial zonal variability in tropical width trends, indicating contraction over the eastern Pacific but expansion in other regions, which amounts to a total narrowing in the zonal mean.

Additionally, the subtropical jet streams have been examined in conjunction with tropical width. Archer and Caldeira (2008) computed integrated properties of the subtropical jet streams and found a poleward movement and weakening of the STJ in both hemispheres, which likely affects the formation of storms in the mid-latitudes and tropics. Manney and Hegglin (2018) examined trends in the subtropical and eddy-driven jet streams, which exhibited strong regional and seasonal differences. Over Africa, a robust tropical widening was observed between the subtropical jets in most seasons, while the STJ of both hemispheres converged towards each other over the eastern Pacific in boreal winter. The southern hemisphere eddy-driven jet exhibited a robust poleward shift, while the NH eddy-driven jet was found to shift equatorward in most seasons and regions. A more recent study by Woollings et al. (2023) found evidence of poleward shifts in both the polar and subtropical jets, which are likely linked to anthropogenic greenhouse gas forcing and stratospheric ozone loss, as well as increasing poleward eddy heat and momentum fluxes, which push the jets poleward. It is still under discussion to what extent the jet location relates to other metrics of tropical width; for example, the poleward edge of the Hadley cells has been found to relate more closely to the eddy-driven jet than to the subtropical jet (Davis and Birner, 2017; Solomon et al., 2016; Waugh et al., 2018), since midlatitude baroclinic eddies reinforce the Hadley cell downwelling and directly give rise to the eddy-driven jet (e. g., Chemke

and Polvani, 2019). The subtropical jet, on the other hand, is rather influenced by tropical processes (Shaw and Tan, 2018) and the latitudinal temperature gradient (Davis and Birner, 2017; Menzel et al., 2019).

To the author's knowledge, analyses of tropical tropospheric width have not been carried out for the PVG tropopause, which combines the tropopause break with subtropical jet latitudes and therefore has the potential to consolidate these different definitions of tropical width, whilst being more closely related to STE transport barriers than conventional tropopause definitions. Therefore, the climatology, trends and variability of the PV-gradient tropopause could provide valuable insights into properties and changes of UTLS transport.

In this paper, we calculate the PVG tropopause for different meteorological reanalyses and investigate its climatology, variability on seasonal to inter-annual time scales and long-term trends. The specific research questions for this paper are: (i) How robust is the representation of the PVG tropopause in different meteorological reanalyses? (ii) What are the dominant modes of variability for the PVG tropopause? (iii) How is the PVG tropopause changing on longer time scales and at different levels, how do these changes translate into changes in upper tropospheric width and how do they relate to tropical width changes?

The data and methods used are described in Sect. 2. Thereafter, Sect. 3 presents and discusses the results, divided into subsections concerning climatological characteristics, seasonal to inter-annual variability, and long-term trends. The final conclusions are presented in Sect. 4.

## 2   Data and methods

The PV gradient-based (PVG) dynamical tropopause (Kunz et al., 2011) is determined as a contour on surfaces of equal potential temperature (i. e., *isentropes*) from the meridional gradient of potential vorticity (PV), combined with the location of the subtropical jet streams. This study is based on four different meteorological reanalyses: ERA-Interim, ERA5, MERRA-2 and JRA-55, which are described in Section 2.1. From six-hourly datasets of potential vorticity ($PV$), potential temperature ($\theta$), zonal and meridional wind speeds ($u$, $v$), we compute the PVG tropopause as explained in Section 2.2. The PVG tropopause is compared to the WMO thermal definition (Section 2.3). Variability and trends of the tropopause are examined via multilinear regression, as detailed in Section 2.4. To reduce computational effort and improve usability, we explore simplifications of the method, which are explained in Section 2.5. We conclude with a regional analysis of the PVG tropopause climatology and trends, which is described in Section 2.6.

### 2.1   Reanalysis datasets

We calculate the PVG tropopause from four recent atmospheric reanalysis datasets which span the global troposphere and stratosphere: ERA-Interim and ERA5 produced by the European Centre for Medium-Range Weather Forecasts (ECMWF), MERRA-2 from the National Aeronautics and Space Administration (NASA), and JRA-55 produced by the Japan Meteorological Agency (JMA). All of these datasets are full-input reanalyses, meaning they combine observations from satellites as

well as from surface stations and upper air measurements. These observations are successively blended with short-range model forecasts in an assimilation scheme, generating best estimates of meteorological parameters in the past.

ERA5 is the most recent reanalysis produced by ECMWF, succeeding ERA-Interim in 2019 (Hersbach et al., 2020). ERA5 is generated by means of a four-dimensional variational assimilation (4D-Var) scheme (Courtier et al., 1994) in conjunction with the Integrated Forecasting System (IFS) cycle 41r2, version 2016. ERA5 extends from 1950 to the present day, providing hourly estimates. The spatial grid has a finer resolution and reaches up to higher vertical levels than ERA-Interim, with a horizontal grid spacing of 31 km ($T_L639$); in the vertical, 137 levels span from the surface to the top level of $0.01\,\mathrm{hPa}$ (Hersbach et al., 2020). We use the updated version ERA5.1, where a low temperature bias in the lower stratosphere has been corrected in the data between 2000 and 2006 (Simmons et al., 2020b). For comparison purposes, we also consider the following, older reanalyses.

ERA-Interim precedes ERA5 and was initially introduced by ECMWF in 2008 (Dee et al., 2011). Constructed through the IFS model cycle 31r2, version 2007, and a four-dimensional variational assimilation (4D-Var) scheme, this dataset covers the satellite era from 1979 to 2019 with six-hourly analysis time steps. ERA-Interim adopted the linear reduced Gaussian grid N128 (wavenumber truncation $T_L255$) with a horizontal grid spacing of approximately $79\,\mathrm{km}$. It comprises 60 vertical levels in hybrid $\sigma$–$p$ coordinates, extending from the surface to the top level at $0.1\,\mathrm{hPa}$ (Fujiwara et al., 2017).

The Japanese 55-year Reanalysis (JRA-55) was launched by the Japan Meteorological Agency in 2013 (Kobayashi et al., 2015); it extends from the start of global radiosonde observations in 1958 to the end of January 2024 and has been succeeded by the reanalysis JRA-3Q (Kosaka et al., 2024). JRA-55 was computed with JMA's operational data assimilation system and global spectral forecast model (GSM) from December 2009, comprising six-hourly datasets. Horizontally, a linear reduced Gaussian grid (N160, equivalent to TL319) was employed with a spacing of $55\,\mathrm{km}$. Vertically, the dataset encompasses 60 hybrid $\sigma$–$p$ vertical levels extending from the surface up to $0.1\,\mathrm{hPa}$ (Kobayashi et al., 2015; Fujiwara et al., 2017).

The Modern-Era Retrospective Analysis for Research and Applications, Version 2 (MERRA-2), introduced by NASA's Global Modeling and Assimilation Office (GMAO) in 2015, provides six-hourly datasets spanning the satellite era from 1980 to the present (Gelaro et al., 2017). Data assimilation is conducted using the Goddard Earth Observing System Model, version 5.12.4 (2015), and a 3D-FGAT ("first guess at the appropriate time") assimilation system (Lawless, 2010; Fujiwara et al., 2017). For its spatial representation, MERRA-2 employs a regular latitude-longitude grid with a spacing of $0.5°$ latitude and $0.625°$ longitude. The vertical grid comprises 72 $\sigma$–$p$ levels, ranging from the surface up to $0.01\,\mathrm{hPa}$ (Fujiwara et al., 2017).

The vertical spacing of the native model grids around the tropopause is approximately $1\,\mathrm{km}$ for ERA-Interim, JRA-55 and MERRA-2 and $0.3\,\mathrm{km}$ for ERA5, transformed from pressure to log-pressure altitude (Fujiwara et al. (2022), Fig. 2.1). For this study, all reanalysis datasets have been interpolated vertically from the native model levels to potential temperature surfaces ($\theta$), using pressure and temperature fields. The reanalysis data is interpolated in steps of $10\,\mathrm{K}$ for ERA-Interim, MERRA-2 and JRA-55, while the finer resolved ERA5 is interpolated in $5\,\mathrm{K}$ steps. ERA5 data is used with a $1° \times 1°$ latitude-longitude grid as provided by the ECMWF, to reduce computational and data storage effort. For the climatological studies, as presented here, this reduced horizontal grid spacing likely has a negligible effect. The PVG tropopause is calculated based on the zonal and meridional wind speeds ($u$ and $v$), potential vorticity ($PV$) and potential temperature ($\theta$) from six-hourly reanalysis data. A

computationally faster alternative using monthly climatologies of reanalyses is detailed in Section 2.5. We consider the period from 1980 to 2017, as it provides the largest overlap of available post-processed reanalysis datasets at the time of this study.

## 2.2 PVG tropopause determination

This section summarizes the underlying dynamical concepts and determination method of the PVG tropopause as established by Kunz et al. (2011). Our contribution applies the methodology to four different reanalyses and incorporates additional criteria to mitigate outliers and noise. The location of the PVG tropopause is calculated on *isentropes*, i. e., surfaces of constant potential temperature. Under the hydrostatic assumption, PV can be expressed depending on potential temperature $\theta$ as a vertical coordinate (e. g., Gettelman et al., 2011), as

$$PV = N^2 \frac{\theta}{\rho g} \left( \zeta_\theta + f \right) . \tag{1}$$

Here, $\rho$ denotes the density, $g$ the gravitational acceleration, and $f$ the Coriolis parameter which represents Earth's rotation-induced vorticity. $N$ refers to the Brunt-Väisälä frequency, and its square is called the buoyancy frequency

$$N^2 = \frac{g}{T} \left( \Gamma_d - \Gamma \right) \tag{2}$$

with the temperature $T$, the temperature lapse rate $\Gamma = -\frac{\partial T}{\partial z}$, wherein $z$ denotes the altitude, and the dry adiabatic lapse rate $\Gamma_d$. The buoyancy frequency is a measure of static stability; if $N^2$ is positive, the ambient air is stably stratified. Transitioning from the troposphere to the stratosphere, static stability (as represented by $N^2$) rapidly increases, which contributes to pronounced meridional PV gradients across the tropopause (see Eq. 1). The variable $\zeta_\theta$ refers to the relative isentropic vorticity, i. e., the vertical component of the rotation of the horizontal wind speed $\boldsymbol{v} = (u, v, 0)$ calculated on an isentropic plane (Hoskins et al., 1985)

$$\zeta_\theta = \left( \frac{\partial v}{\partial x} \right)_\theta - \left( \frac{\partial u}{\partial y} \right)_\theta . \tag{3}$$

As apparent from Eq. 3, relative vorticity is based on gradients of zonal and meridional wind speed. In regions of strong horizontal wind shear—e. g., at the flanks of the subtropical jets—$\zeta_\theta$ changes noticeably, which further enhances meridional PV gradients near the subtropical jets and tropopause break. In summary, both the rapid changes of static stability across the tropopause, as well as the strong horizontal wind shear at the subtropical jet streams contribute to steep PV gradients near the tropical tropopause break, laying the foundation for the PV-gradient tropopause in this region.

On each isentrope, horizontal coordinates of *equivalent latitude* are introduced. Equivalent latitude ($\varphi_e$) is a one-to-one mapping of potential vorticity, determined from contours of equal PV on an isentropic surface (Butchart and Remsberg, 1986). For each PV value between -30 and +30 PVU in steps of 0.1 PVU, the corresponding PV isoline is found, and the area $A(PV)$ polewards of this isoline is computed. A circle with the same area $A(PV)$ is centered at the North or South Pole, respectively, and the radius of this circle in degrees of latitude is defined as the equivalent latitude

$$\varphi_e = \arcsin \left( 1 - \frac{A(PV)}{2\pi R_E^2} \right) , \tag{4}$$

where $R_E$ is the Earth's radius. On each isentrope, we determine the equivalent latitude of the tropopause break, $\varphi_e^{TP}(\theta)$, from the potential vorticity gradient and horizontal wind speed. For this, the potential vorticity gradient is computed numerically as the derivative of the preset PV values with respect to equivalent latitude $\frac{\partial PV(\varphi_e, \theta)}{\partial \varphi_e}$. As the subtropical jet streams correlate with sharp PV changes across the tropopause break, the highest PV gradient in the vicinity of the subtropical jet stream is searched for the determination of the PVG tropopause. To account for the subtropical jets, we calculate the horizontal wind velocity from zonal ($u$) and meridional wind speeds ($v$) in each reanalysis. First, zonal and meridional winds are averaged along each PV contour in order to obtain the mean along equivalent latitudes, from here on called the *equivalent latitude zonal mean*. The average horizontal wind speed at each equivalent latitude is calculated from the average zonal and meridional winds as $v_h = \sqrt{u^2 + v^2}$. In order to also obtain the conventional latitude $\phi$ as a coordinate, the latitude values along each PV contour are averaged similarly to the winds. As at the tropopause break, strong horizontal winds and sharp PV gradients occur, we multiply the PV gradient with the horizontal wind velocity, defining this product as the quantity $Q$:

$$Q(\varphi_e, \theta) = \frac{\partial PV(\varphi_e, \theta)}{\partial \varphi_e} v_h(\varphi_e, \theta). \tag{5}$$

The algorithm then searches for maxima of $Q$. Therefore, the PVG tropopause break on each isentrope $\theta$ is defined as the equivalent latitude where the highest local maximum of $Q$ occurs, i.e.,:

$$\varphi_e^{TP}(\theta) = \varphi_e(\theta)|_{\max(Q)}, \tag{6}$$

upon certain conditions for the maxima, which are described below. For a visual representation of the PVG tropopause break on certain isentropes, refer to Kunz et al. (2011). In this method, the tropopause break was searched in an equivalent latitude range of $5° < |\varphi_e| < 85°$ in each hemisphere, and is defined as the absolute maximum of $Q$. However, in seasons with a strong polar jet—especially during austral winter—the $Q$ maximum corresponding to the polar jet occasionally exceeds that of the subtropical jet and is falsely recognized by the algorithm as the tropical tropopause break. As a consequence, the method is altered to preferentially select the subtropical jet maximum, such that multiple local maxima are detected and the most equatorward maximum is chosen as the PVG tropopause. In order to exclude the polar jets, the maxima search is empirically confined to a poleward limit in equivalent latitude of $\pm 80°$ between the 320 K and 350 K isentropes, and $\pm 55°$ above 350 K. Furthermore, the minimum equivalent latitude is set to $\pm 15°$ in order to exclude equatorward outliers. The equivalent latitude intervals in this study are therefore $15° < |\varphi_e| < 80°$ between 320 K and 350 K, and $15° < |\varphi_e| < 55°$ above 350 K. To ensure that only substantial maxima in $Q$ are chosen, a threshold is set for the *prominence* of the maxima. The prominence is defined as the height difference between a local maximum and the highest local minimum next to it. In the search for $Q$ maxima, the relative prominence is set to amount to at least 0.2 times the absolute maximum of $Q$ on each isentrope.

On every isentropic level ranging from 320 K to 380 K and for each time step in the original reanalysis datasets, a comprehensive profile of the variables relevant to the PVG tropopause is computed. This profile includes latitude ($\phi$), the combined variable ($Q$), potential vorticity ($PV$), zonal wind speed ($u$), and horizontal wind speed ($v_h$). All these variables are averaged over potential vorticity contours and are dependent on the equivalent latitude ($\varphi_e$) as a dimension. Time series for the PVG tropopause are then generated for each isentropic level. These time series encapsulate the values of the aforementioned vari-

ables at the PVG tropopause, specifically the equivalent latitude ($\varphi_e^{TP}$), potential vorticity ($PV^{TP}$), zonal wind speed ($u^{TP}$), horizontal wind speed ($v_h^{TP}$), and mean latitude ($\phi^{TP}$).

## 2.3 WMO lapse-rate tropopause

For the purpose of comparing the PVG tropopause with conventional tropopause definitions, the thermal tropopause has been determined according to the World Meteorological Organization (WMO; World Meteorological Organization, 1957). The WMO thermal tropopause marks the rapid increase of static stability $N^2$ that occurs at the transition from troposphere to stratosphere (see Eq. 2), and is based on characteristics of the temperature lapse rate $\Gamma$. More precisely, the WMO thermal tropopause is defined as the lowest altitude where $\Gamma$ falls below $2\,\mathrm{K\,km^{-1}}$, provided the average lapse rate between this level and all higher levels within $2\,\mathrm{km}$ remains below $2\,\mathrm{K\,km^{-1}}$. In our study, the WMO tropopause is determined from the temperature in each reanalysis dataset. Transformed to potential temperature coordinates, this yields the potential temperature of the WMO tropopause dependent on latitude, $\theta(\phi)$.

## 2.4 Determination of variability and trends

In order to disentangle long-term changes and variability in the PVG tropopause such as the seasonal cycle, El Niño Southern Oscillation (ENSO) and Quasi Biennial Oscillation (QBO), we apply a multilinear regression method to the time series of tropopause latitude and potential vorticity on each isentropic level. The monthly mean time series for each hemisphere of tropopause latitude $\phi_\theta^{TP}(t)$ and potential vorticity $PV_\theta^{TP}(t)$ (generalized as $f$ in Eq. 7), for the latitude also the hemispheric difference (NH-SH), are fit to the multilinear regression model in Eq. 7 using least-squares approximation. Regressors include the time $t$ for long-term trends, the mean seasonal cycle $S$, two orthogonal QBO indices represented by the monthly mean zonal wind speeds near the equator on pressure levels $30\,\mathrm{hPa}$ ($QBO_{30}$) and $50\,\mathrm{hPa}$ ($QBO_{50}$) following Stiller et al. (2012), and the ENSO index $ENSO$

$$f_\theta^{TP}(t) = a_0 + a_{lin} \cdot t + a_{seas} \cdot S(t) + a_{qbo30} \cdot QBO_{30}(t) + a_{qbo50} \cdot QBO_{50}(t) + a_{enso} \cdot ENSO(t) + R(t). \tag{7}$$

Here, $R(t)$ denotes the residual of the fit and $a_0$ the vertical offset. The regressors $S$, $QBO_{30}$, $QBO_{50}$ and $ENSO$ are determined in advance of the multilinear fit: The mean seasonal cycle $S(t)$ is computed directly from the monthly mean time series by averaging each monthly value over the climatological period. Regarding the QBO, two orthogonal indices are considered, i. e., monthly mean zonal wind speeds near the equator on isobars $30\,\mathrm{hPa}$ and $50\,\mathrm{hPa}$, which have been observed at Singapore weather station ($1°$ N, $104°$ E) (Naujokat, 1986). The bi-monthly multivariate ENSO index, MEI.v2, combines five variables: sea level pressure, sea surface temperature, zonal and meridional surface wind speeds, as well as outgoing longwave radiation, in the tropical Pacific region ($30°$ N $- 30°$ S and $100°$ E $- 70°$ W). MEI.v2 is provided by the National Oceanic and Atmospheric Administration (NOAA) (Zhang et al., 2019).

To ensure that the regression coefficients $a_i$ yield the amplitudes of each variability, the regressors $S$, $QBO_{30}$, $QBO_{50}$ and $ENSO$—furthermore generalized as $y(t)$—are normalized before fitting, i. e., divided by their maximum amplitude, which is calculated as half the difference between absolute maximum and minimum of the regressor $(y(t)_{max} - y(t)_{min})/2$. The

amplitude $a_{qbo}$ of the combined QBO variability factor, i. e., on both $30\,\mathrm{hPa}$ and $50\,\mathrm{hPa}$, is computed as follows: First, the time series of latitude/PV are fitted to the $30\,\mathrm{hPa}$ and $50\,\mathrm{hPa}$ QBO indices separately, as indicated in Eq. 7. The two resulting terms are added $f_{qbo}(t) = a_{qbo30} \cdot QBO_{30}(t) + a_{qbo50} \cdot QBO_{50}(t)$, and the maximum amplitude $a_{qbo}$ is computed as

$$290 \quad a_{qbo} = \frac{1}{2}\left(\max(f_{qbo}) - \min(f_{qbo})\right). \tag{8}$$

## 2.5 Simplifications of the method

Since determining the PVG tropopause from sub-daily global reanalysis data is computationally expensive, we test simplifications of the algorithm to enhance usability. Specifically, we examine the influence of using monthly or zonal climatologies of reanalysis data and omitting the subtropical jet wind criterion. The following additional analyses are carried out using ERA5 data and compared with the standard method (i.e., six-hourly, global reanalysis data with wind criterion):

1. Monthly reanalysis climatologies

2. Monthly reanalysis climatologies without the wind criterion

3. Monthly and zonal mean reanalysis climatologies

4. Monthly and zonal mean reanalysis climatologies without the wind criterion

For analyses (3) and (4), ERA5 data are also averaged over longitude. In these cases, the computation of equivalent latitude is omitted, and the meridional gradient of PV is computed with respect to geographical latitude. In analyses (2) and (4), the wind criterion is omitted, and only the PV gradient is computed, excluding $v_h$ from Eq. 5. To compare the results, the seasonal multi-year climatologies, mean seasonal cycle, and trends computed with aforementioned methods are discussed in Section 3.5.

## 2.6 Regional aspects

Several studies indicate strong regional variations of long-term trends in tropical width, specifically the tropopause break (Martin et al., 2020) and the subtropical jets (Manney and Hegglin, 2018). Therefore, a similar regional analysis of the PVG tropopause break latitude is conducted here, using ERA5 reanalysis data in the time range from 1980–2017. Since the PVG tropopause is computed using equivalent latitude, which directly corresponds to PV, the tropopause on each isentrope aligns with a specific PV value. By identifying this PV contour on each isentrope within reanalysis data, we can map the latitude-longitude contour of the tropopause on each isentropic surface.

These PV contours are used to create global tropopause surfaces $\theta^{TP}(\phi, \lambda)$. In order to obtain the lowest potential temperature for the tropopause, we iterate through the isentropes from top to bottom, starting with the highest potential temperature level of $380\,\mathrm{K}$ down to $320\,\mathrm{K}$ in steps of $5\,\mathrm{K}$. Within the loop, the PV field on each isentrope is obtained from the reanalysis. As PV fields can be noisy, outliers are masked using a $3\sigma$ filter which iterates through latitude, computes the average $\mu$ and standard deviation $\sigma$ of PV within each latitude circle, and masks all PV values outside of the interval $[\mu - 3\sigma, \mu + 3\sigma]$ such that these outliers are ignored in the subsequent computation. The tropopause contour on each isentrope is found by

searching for the corresponding value $PV^{TP}$ in the PV field. The geographical area where PV is more extreme than at the tropopause, $|PV| \geq |PV^{TP}|$, is then set to the current potential temperature $\theta$. This procedure is repeated until reaching the lowest isentrope, 320 K.

Originally, the PVG tropopause is only defined near the tropopause break and the subtropical jet. In order to create a global tropopause approximating a physical transport barrier, we extend the tropopause in the extratropics with a PV isosurface, similarly to the traditional dynamical tropopause: Beyond the 320 K contour—e. g., in the area where PV is more extreme than at the 320 K tropopause—, we continue the tropopause with the isosurface corresponding to PV at the tropopause on 320 K. This PV surface is obtained from a piecewise-linear 1D interpolation of potential temperature over PV at each point, i. e., $\theta(\phi, \lambda, PV)$.

The result of this iterative algorithm is a global field of tropopause height $\theta^{TP}(\phi, \lambda)$. Lastly, a median filter with a circular kernel of $4°$ radius is run over the whole field in order to reduce noise and outliers.

To examine regional long-term trends, the PVG tropopause contour is determined on each isentrope from monthly mean ERA5 data as detailed above. The intersections of the tropopause with every tenth meridian are saved as timeseries $\phi^{TP}(\lambda, \theta, t)$ and analyzed using multilinear regression as described in Section 2.4.

## 3   Results

This section presents the climatology, variability and long-term changes of the PVG tropopause and compares this definition to traditional tropopause definitions such as the WMO lapse rate tropopause and the 2 PVU surface. The robustness of the PVG tropopause is assessed in four different reanalyses: ERA5, ERA-Interim, MERRA-2 and JRA-55. Finally, variability analysis aims to disentangle the effects of the seasonal cycle, El Niño Southern Oscillation, Quasi-Biennial Oscillation, and a long-term trend on the PVG tropopause location and potential vorticity.

### 3.1   Climatological structure of the PVG tropopause

The climatological structure of the PVG tropopause is examined based on 1980–2017 climatologies of the tropopause latitude and potential vorticity in winter and summer of each hemisphere. To investigate the location of the tropopause in comparison to the PV distribution and subtropical jets following Kunz et al. (2011), Fig. 1 illustrates seasonal climatologies of the PVG tropopause plotted as a function of latitude and potential temperature, along with the WMO thermal tropopause. All shown quantities are calculated as zonal means in an equivalent latitude–based coordinate system and subsequently transformed into actual latitudes, as described in the previous section. Since the results are similar for all four considered reanalyses, only the most recent dataset, ERA5, is visualized and discussed in this section. A detailed comparison of the results for different reanalyses and a discussion of robustness is presented in Sect. 3.3.

According to its definition, the estimated PVG tropopause coincides with the regions of highest PV gradient, visible as the areas with highest PV contour density in Fig. 1, as well as to the $u$ maxima representing the subtropical jet cores. In agreement with Kunz et al. (2011), the dynamical tropopause is not well represented by any constant PV value, but instead crosses several

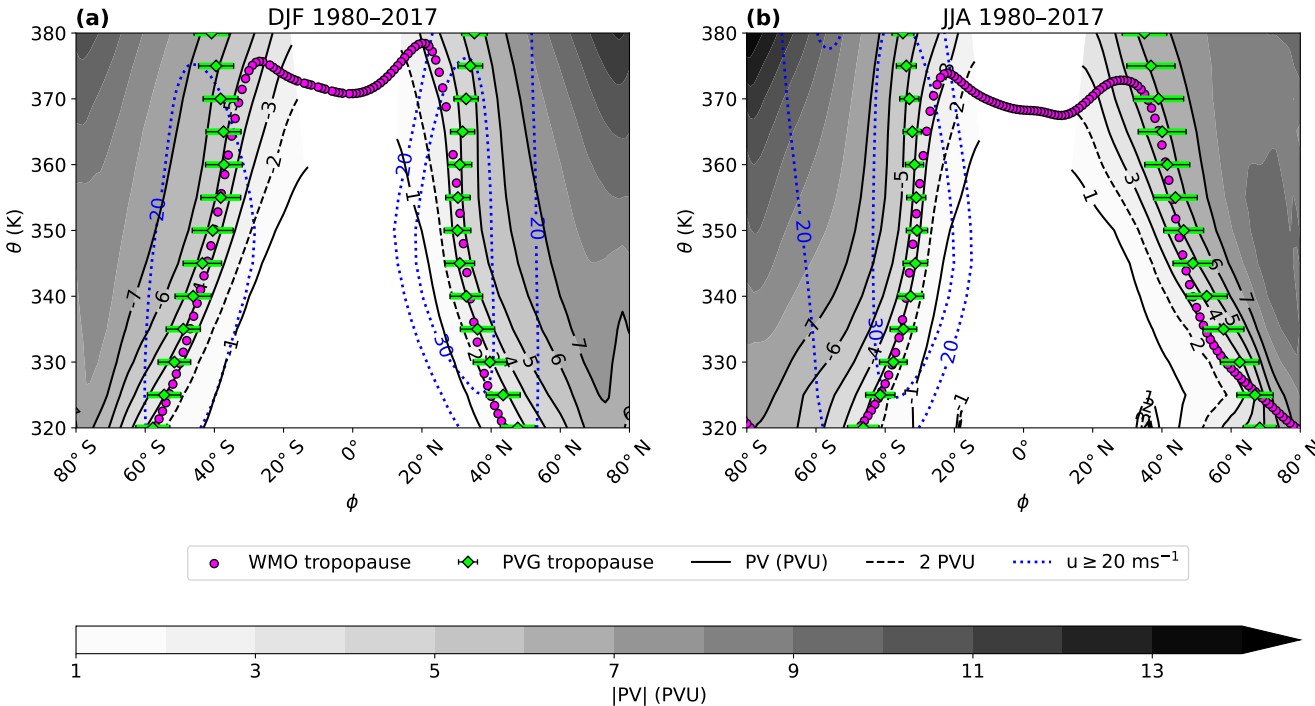

**Figure 1.** Seasonal climatology in **(a)** December–February (DJF) and **(b)** June–August (JJA) 1980–2017 of the PVG tropopause latitude $\phi^{TP}$ averaged over equivalent latitude contours, compared to the WMO lapse-rate tropopause in front of the PV-field (grayscales with solid black PV isolines between $\pm 1$ and $\pm 7$ PVU) and the zonal wind $u$ maxima indicating the locating of the subtropical jets (dotted blue contours), displayed in the latitude/potential temperature plane. A similar figure has been published by Kunz et al. (2011) for ERA-Interim and is recreated here in a slightly altered form for ERA5.

PV isolines, while PV tends to increase with potential temperature (see Eq. 1). Hence, the common practice of defining the

350 dynamical tropopause as a PV isosurface (e. g., 2 PVU) for studies of stratosphere-troposphere exchange (Holton et al., 1995), is generally an oversimplification. Figure 1 further shows that the PVG tropopause agrees well with the WMO lapse rate tropopause up to ~360 K throughout different seasons and hemispheres.

Comparison of the December–February climatology in Fig. 1a to the June–August climatology in Fig. 1b exhibits a distinct seasonal cycle in the PVG tropopause, characterized by substantial shifts into the summer hemisphere due to the seasonal ITCZ

355 movement. As observed in Fig. 1b, this poleward shift in summer is especially prominent in the northern hemisphere between the 320 K to 340 K surfaces, where the PVG tropopause extends far poleward until 70° N, while in austral summer (SH in Fig. 1a), the PVG tropopause reaches only until 60° S on 320 K. We hypothesize that these differences in the PVG tropopause latitude between austral and boreal summer stem from the seasonal variations of the subtropical and eddy-driven jet streams. In the summer of both hemispheres, the subtropical jets weaken (Gettelman et al., 2011), which is evident regarding the $u$

360 contours in Fig. 1, exhibiting substantially lower wind speeds in the summer hemisphere (i. e., SH in Fig. 1a, NH in Fig. 1b)

than in the winter hemisphere. Additionally, the subtropical jet often coalesces with the eddy-driven jet (e. g., Manney and Hegglin, 2018). Especially during phases of lower jet wind speeds in summer, the subtropical and eddy-driven jets merge at lower levels, resulting in a poleward displaced PVG tropopause. Also during summer at the lower isentropes, the poleward displaced PVG tropopause agrees well with the thermal tropopause. In boreal summer, as shown in the right half of Fig. 1b, the subtropical jet is even weaker than in austral summer (left half of Fig. 1a). Additionally, Manney and Hegglin (2018) show that the eddy-driven jet extends farther poleward in boreal summer than in austral summer, reaching latitudes northward of $70°$ N, which corresponds to the high latitudes of the PVG tropopause observed at $320$ K–$330$ K in Fig. 1b. In brief, the weakening of the subtropical jet in summer, coalescence of subtropical and eddy-driven jets, and farther poleward extension of the NH eddy-driven jet likely attributes to the far poleward displacement of the tropopause in boreal summer. Furthermore, the variability of the PVG tropopause (measured in terms of standard deviation) is larger in summer than in winter of each hemisphere, which is probably due to the aforementioned subtropical jet weakening in summer, creating wider, less pronounced wind maxima and therefore larger fluctuations in the PVG tropopause (Eq. 5).

Figure 1 reveals that the PVG tropopause corresponds well with the WMO definition up to about $360$ K, with the thermal tropopause largely ranging within the standard deviation of the PVG tropopause. During boreal summer (right half of Fig. 1b), the thermal tropopause is located somewhat more poleward on the $320$ K isentrope and lies above the PVG tropopause between $60°$ N and $80°$ N. This weak deviation of thermal and PVG tropopause is possibly caused by cyclonic systems in the upper troposphere, which often shift the 2 PVU surface below the thermal tropopause. As the PVG tropopause closely corresponds to the 2 PVU surface on lower isentropes, this could explain the different locations of both tropopauses there (Kunz et al., 2011; Wirth, 2000, 2001).

Above $360$ K, the thermal WMO tropopause narrows until reaching maximum heights of $370$ K to $380$ K between $20°$ to $40°$ latitude, and drops slightly near the equator. The PVG tropopause, on the other hand, narrows to $30°$ latitude around the $350$ K isentrope and widens poleward above, reflecting variations in the location of maximum PV gradients and jet streams. As the PVG tropopause corresponds both to the PV gradient and to zonal wind ($u$) maxima of the subtropical jets, lower PV gradients lead to a stronger influence of $u$ maxima, causing the PVG tropopause to follow the subtropical jets more closely on higher isentropes—this can be observed in both panels of Figure 1, where the PVG tropopause coincides with the regions of maximum PV gradient up to $360$ K to $370$ K, recognizable by closely spaced PV isolines; while above, the PVG tropopause tends to align with the horizontal wind contours. As the subtropical jet streams mark a transport barrier, this relation to the PVG tropopause is an asset in transport studies, providing a valuable complement to other definitions.

Near the equator, however, the PVG tropopause cannot be computed, as PV is ill-defined there. Additionally, PV gradients and jet streams weaken above $380$ K, leading to weak $Q$ maxima (see Eq. 5) which frequently results in an undefined PVG tropopause. Below $380$ K, the PVG tropopause is mostly well-defined. As a consequence, the PVG tropopause can be considered a reliable definition up to $370$ K to $380$ K in the subtropics and midlatitudes.

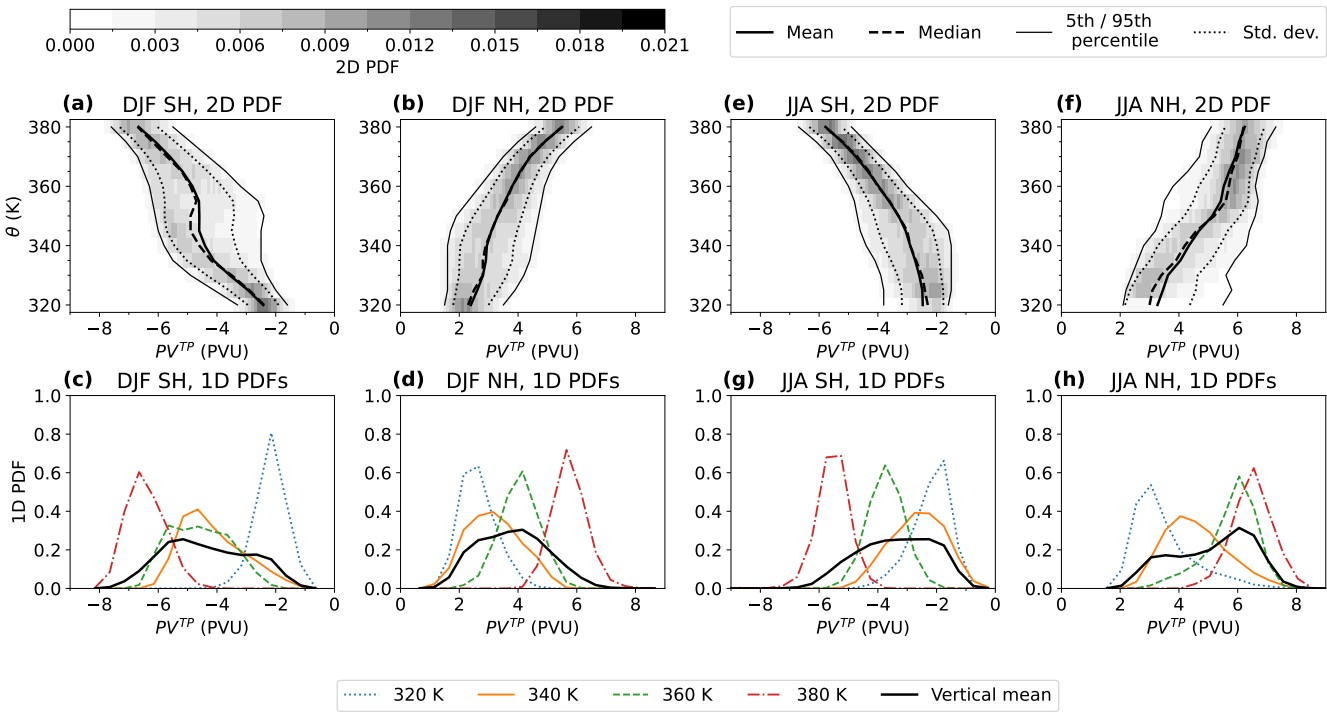

**Figure 2.** Seasonal climatology of the PV distribution at the PVG tropopause $PV^{TP}$ as probability density functions (PDFs) for ERA5 between 1980 and 2017, following Kunz et al. (2011). The left-hand side (panels **a–d**) shows the December–February (DJF) climatology, while the right-hand side (panels **e–h**) presents June–August (JJA). The upper panels (**a, b, e, f**) display the 2D PDF in the PV, $\theta$ plane to illustrate the change of the PV distribution with height, with a bin size of 0.1 PVU and 5 K for ERA5, 10 K for all other reanalyses. Additionally, the mean, median, $1\sigma$ interval, 5th and 95th percentile of the PDFs are delineated. In the lower panels (**c, d, g, h**), the corresponding 1D PDFs on selected isentropes are drawn as coloured lines. The vertical mean 1D PDF along the whole tropopause across 320 K to 380 K is specified as a heavy black line.

## 3.2 Potential vorticity at the PVG tropopause

This section examines the potential vorticity $PV^{TP}$ at the PVG tropopause, and how $PV^{TP}$ varies between hemispheres and
seasons. To visualize in which range and frequency PV occurs at the tropopause, Fig. 2 displays probability density functions (PDFs) of $PV^{TP}$ between 1980 and 2017. As an example, results for the newest reanalysis ERA5 are shown here; comparisons between different reanalyses can be found in Sect. 3.3.

On the left-hand side of Fig. 2, panels (a–d) display the PV climatologies of December–February (DJF), while the panels (e–h) on the right-hand side present the season June–August (JJA). To this end, the PVG tropopause and corresponding PV
values are determined through the method described in Sect. 2.2 on each isentropic level between 320 K and 380 K for four

times daily from 1980 to 2017. The results are then binned into each of the four seasons, of which only DJF and JJA are shown here. From each seasonal dataset of $PV^{TP}$, the probability density functions are computed.

The upper row in Fig. 2 (panels a, b, e, f) displays two-dimensional PDFs in the PV–$\theta$ plane, which are calculated as follows: For each hemisphere, $PV^{TP}$ is binned into two-dimensional histograms with a bin size of $0.1\,\mathrm{PVU}$ horizontally and
$5\,\mathrm{K}$ vertically for ERA5, $10\,\mathrm{K}$ for other reanalyses. These histograms are normalized to compute the two-dimensional PDFs. In order to illustrate further statistical properties of the PV distributions, Fig. 2 (a, b, e, f) delineates the means, medians, standard deviations, as well as 5th and 95th percentiles on top of the 2D PDFs.

To highlight the $PV^{TP}$ distribution on selected isentropes, one-dimensional PDFs are shown in the bottom row of Fig. 2 (panels c, d, g and h) which are computed as one-dimensional normalized histograms of $PV^{TP}$ with bin sizes of $0.5\,\mathrm{PVU}$
on each isentropic level. In order to visualize the PV distribution along the tropopause across several isentropes, a vertically averaged 1D PDF is computed as the arithmetic mean of all 1D histograms between $320\,\mathrm{K}$ and $380\,\mathrm{K}$.

The two-dimensional PDFs for December–February in Fig. 2a and 2b indicate that between $320\,\mathrm{K}$ and $380\,\mathrm{K}$, the absolute value of potential vorticity at the tropopause, $|PV^{TP}|$, increases with height. Extremes of $PV^{TP}$ reach $\pm1.5\,\mathrm{PVU}$ at $320\,\mathrm{K}$ to $\pm8\,\mathrm{PVU}$ at $380\,\mathrm{K}$. The variability, outlined by the standard deviation and 5th/ 95th percentiles, amounts to $1\,\mathrm{PVU}$ to $2\,\mathrm{PVU}$,
maximizing between the isentropes of $340\,\mathrm{K}$ to $360\,\mathrm{K}$. In austral summer (Fig. 2a), the variability tends to be larger than in boreal winter (Fig. 2b). The mean and median of $PV^{TP}$ range from $-2\,\mathrm{PVU}$ to $-7\,\mathrm{PVU}$ in austral summer and $2\,\mathrm{PVU}$ to $6\,\mathrm{PVU}$ in boreal winter. Mean and median closely align, except for $340\,\mathrm{K}$ to $350\,\mathrm{K}$ in Fig. 2a where the largest variability occurs.

The one-dimensional PDFs for December–February displayed in Fig. 2c and 2d confirm the increase of $|PV^{TP}|$ with
height. The mode (i. e., peak) of each 1D PDF, which represents the most probable PV value at the tropopause, increases from ca. $\pm2.5\,\mathrm{PVU}$ on the $320\,\mathrm{K}$ isentrope to around $\pm6\,\mathrm{PVU}$ near the tropical tropopause at $380\,\mathrm{K}$. Comparing the 1D PDFs in austral summer (Fig. 2c) to boreal winter (Fig. 2d), the modes are similar on the $320\,\mathrm{K}$ isentropes, but are offset considerably towards larger $|PV^{TP}|$ on isentropes over $320\,\mathrm{K}$. Additionally, the distribution on $320\,\mathrm{K}$ exhibits a taller, sharper peak in austral summer.

Furthermore, the 1D distribution's peak widths reflect the vertical change of variability observed in the 2D PDFs above. Panels 2c and 2d exhibit relatively narrow peaks corresponding to lower variability on $320\,\mathrm{K}$ and $380\,\mathrm{K}$ and wider peaks corresponding to higher variability on $340\,\mathrm{K}$ and $360\,\mathrm{K}$. Regarding the 1D PDF variabilities in austral summer (Fig. 2c) by contrast with boreal winter (Fig. 2d), the distribution on the $360\,\mathrm{K}$ isentrope is substantially wider and exhibits much larger PV variance, reflecting the larger variance observed in the 2D PDF for austral summer (Fig. 2a). The larger PV variance in summer
is likely due to the weakening of the subtropical jet and coalescence with the eddy-driven jet (Manney and Hegglin, 2018).

The vertical mean PV distributions across the tropopause in December–February (Fig. 2c and 2d) exhibit a wide peak structure, which manifests in two peaks in austral summer (Fig. 2c) and only one peak in boreal winter (Fig. 2d). The two-peak structure in austral summer is related to the pronounced shift towards higher $|PV^{TP}|$ on the $340\,\mathrm{K}$, $360\,\mathrm{K}$ and $380\,\mathrm{K}$ isentropes, while the mode at $320\,\mathrm{K}$ stays roughly the same, as described above.

The right-hand side of Fig. 2 (panels e–h) shows the potential vorticity PDFs for June–August. Both the one- and two-dimensional distributions confirm that $|PV^{TP}|$ increases with height. Regarding the 2D PDFs in the upper panels, Fig. 2e and 2f, an overall offset of $PV^{TP}$ towards more positive values can be observed in June–August compared to December–February, with extremes ranging from $-1\,\mathrm{PVU}$ to $-7\,\mathrm{PVU}$ in austral winter (Fig. 2e) and $2\,\mathrm{PVU}$ to $8\,\mathrm{PVU}$ in boreal summer (Fig. 2f). This seasonal offset towards positive PV likely corresponds to the northward shift of the ITCZ in June–August. The variance and distance between the 5th and 95th percentiles are larger in boreal summer (Fig. 2f) than in austral winter (Fig. 2e), which leads to the conclusion that PV variance is generally larger in the summer of each hemisphere than in winter. Moreover, the PV PDFs for the summer of each hemisphere (austral summer: Fig. 2a, boreal summer: Fig. 2f) differ more strongly from each other than the winter seasons (boreal winter: Fig. 2b, austral winter: Fig. 2e), where the PV distributions appear almost symmetric. This larger variability in summer is also apparent from the PVG tropopause climatology in Fig. 1, and is likely related to the weakening of the summer hemisphere STJ, leading to wider $Q$ maxima (see Eq. 5).

The one-dimensional PDFs for June–August shown in the lower right panels, Fig. 2g and 2h, substantiate that the most prominent seasonal changes in potential vorticity occur on the isentropes between $340\,\mathrm{K}$ and $360\,\mathrm{K}$, where the mode of $|PV^{TP}|$ shifts by roughly $2\,\mathrm{PVU}$, as was already observed in the 1D PDFs for December–February (Fig. 2c and Fig. 2d). The large PV changes between $340\,\mathrm{K}$ and $360\,\mathrm{K}$ account for the second peak around $\pm6\,\mathrm{PVU}$ observed in the vertically averaged 1D PDFs in each summer hemisphere, Fig. 2c and Fig. 2h.

To summarize, Fig. 2 provides evidence that the potential vorticity at the tropopause $PV^{TP}$ covers the range $\pm1.5\,\mathrm{PVU}$ to $\pm8\,\mathrm{PVU}$ with negative values in the southern hemisphere and positive values in the northern hemisphere, reaching larger absolute values with height. Seasonal changes include an ITCZ-related offset towards more positive PV during June–August, which is especially prominent on the isentropes $340\,\mathrm{K}$ to $360\,\mathrm{K}$, as well as larger PV variability at the tropopause in the summer hemisphere, which is likely linked to the general variability of the PVG tropopause in summer due to STJ weakening. These findings are similar in the different reanalyses ERA5 (displayed e. g., in Fig. 2), ERA-Interim, MERRA-2 and JRA-55 (not shown here). In agreement with Kunz et al. (2011), our results confirm that potential vorticity spans a wide range across the tropopause, which is not reflected in common dynamical tropopause definitions consisting of a single $1.5\,\mathrm{PVU}$ to $4\,\mathrm{PVU}$ isosurface. The PVG tropopause accounts for this variability in potential vorticity, suggesting an advantage of this definition over the PV-monosurface dynamical tropopause.

### 3.3 Robustness of the PVG tropopause representation

After consideration of the tropopause shape and potential vorticity distributions for the single reanalysis ERA5 in Sects. 3.1 and 3.2, this section examines the robustness of the representation of the PVG tropopause in different meteorological reanalyses (ERA5, ERA-Interim, MERRA-2 and JRA-55). To assess robustness among the reanalyses, the latitudes and potential vorticity distributions of the PVG tropopause are shown in Figs. 3 and 4 as climatological means. The respective figures show the 1980–2017 climatological mean latitude $\phi^{TP}$ and potential vorticity $PV^{TP}$ at the PVG tropopause in the solstice seasons on both hemispheres for all four reanalyses. To visualize the tropopause variability, the standard deviations determined from ERA5 are also shown.

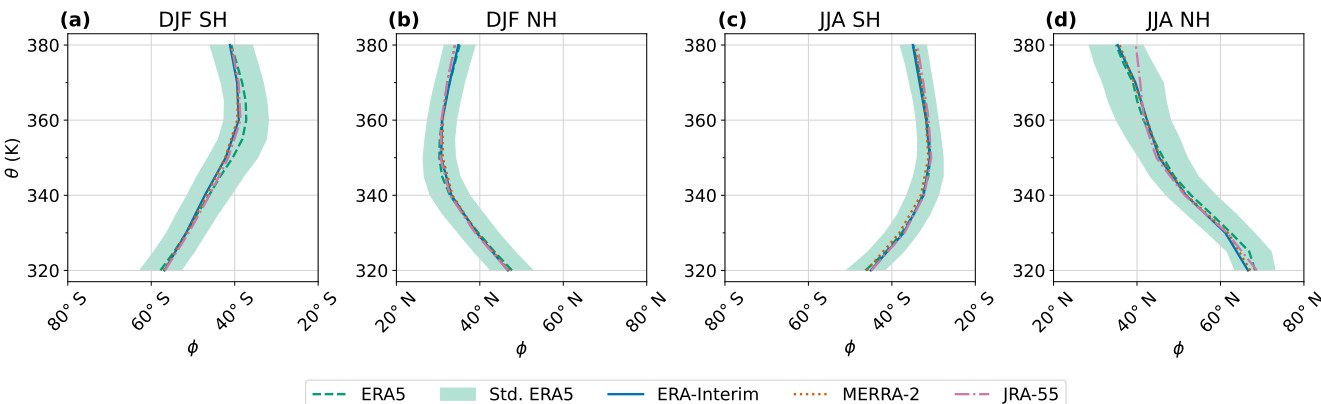

**Figure 3.** Comparison of the PVG tropopause latitude $\phi^{TP}$, averaged over equivalent latitude contours, in four reanalyses. Seasonal climatologies (1980–2017) of $\phi^{TP}$ in each hemisphere are shown for December–February (DJF) in the southern hemisphere **(a)** and northern hemisphere **(b)**; for June–August (JJA) in the southern hemisphere **(c)** and northern hemisphere **(d)**. Lines indicate the mean climatology in each reanalyses, the standard deviation of ERA5 is shaded in green.

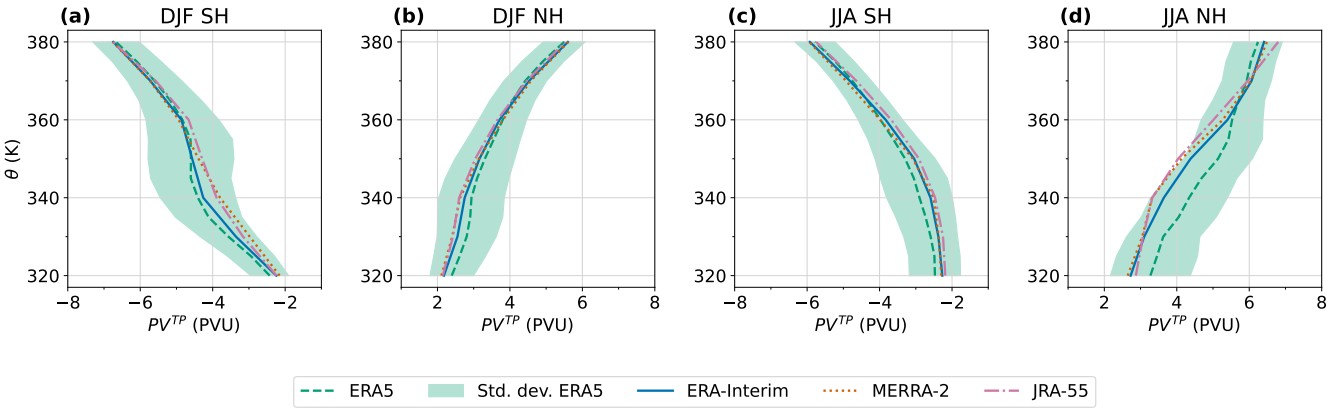

**Figure 4.** Means and standard deviations of the PDFs of $PV^{TP}$ (as shown in the upper panels of Fig. 2) in four reanalyses. Similarly to Fig. 3, the seasonal climatologies between 1980 and 2017 are shown for each hemisphere in December–February (DJF) in panels **(a)** and **(b)**, for June–August (JJA) in panels **(c)** and **(d)**. Lines indicate the $PV^{TP}$ climatological means in each reanalyses; the standard deviation of ERA5 is shaded in green.

Figure 3 reveals that the PVG tropopause latitude $\phi^{TP}$ is consistent among different reanalyses. Moreover, the seasonal variability of the tropopause location, with substantial shifts into the summer hemisphere as observed in Fig. 3a and 3d, turns out to be very similar in the different datasets (discussed in more detail in Section 3.4). In particular, latitudinal differences between different reanalyses are on the order of $1°$, i.e., one order of magnitude smaller than the seasonal variability which is on the order of $10°$. Note that the reanalysis differences are larger in summer (Fig. 3a and 3d), occasionally reaching up to

$3°$, with one exceptionally large deviation in boreal summer (Fig. 3d) at 380 K, where JRA-55 exceeds the other reanalyses by around $5°$. These stronger differences between reanalyses in summer are potentially related to the larger PVG tropopause variability observed in summer, which is likely due to the weakening of the STJ and related widening of $Q$ maxima (see Eq. 5) as described in Sect. 3.1.

Figure 4 manifests the increase of potential vorticity $|PV^{TP}|$ at the PVG tropopause with increasing potential temperature, as observed for ERA5 in Fig. 2, robustly in all four reanalyses, as well as the related seasonal variability. Overall the differences in $PV^{TP}$ between the four reanalyses are smaller than the natural variability reflected in the standard deviation of ERA5 values. However, in some seasons and levels, part of the reanalyses exhibit noticeable differences in the means of $PV^{TP}$. For instance, in austral summer (Fig. 4a) around 340 K, differences between ERA5 and MERRA-2 amount to 0.5 PVU. In boreal summer (Fig. 4d) between 320 K and 350 K, differences between the reanalyses even reach up to 1 PVU. Overall, the PV change with height and seasonality is qualitatively similar in different reanalyses. Quantitatively, however, the PVG tropopause PV values differ between the reanalyses, and are less robust than the tropopause latitude. ERA5 tends to yield the largest, MERRA-2 and JRA-55 the smallest absolute values of $PV^{TP}$. Examination of the potential vorticity fields reveals considerable variation in PV values and PV gradients among the reanalyses. ERA5, for instance, features the most extreme PV values and steepest PV gradient, which contributes to the observed $PV^{TP}$ differences between the reanalyses. However, the latitude of the strongest PV gradient remains similar in all reanalyses. As a result, the latitude of the PVG tropopause $\phi^{TP}$ is more robust than the corresponding PV value $PV^{TP}$.

Given the differences in PV values among the four reanalyses, we advise using the same reanalysis for comparisons between the PVG tropopause and other variables. In particular, the tropopause PV values need to be consistently calculated from one dataset. This mitigates the risk of introducing errors stemming from variations in the PV fields across different reanalyses. However, comparing multiple reanalyses or models is important to represent the range of uncertainty in the state of the atmosphere.

## 3.4 Variability and trends

To examine the variability and long-term changes of the PVG tropopause, we analyze the monthly mean time series of the tropopause latitude $\phi_\theta^{TP}(t)$ and potential vorticity $PV_\theta^{TP}(t)$ between 1980 and 2017 on every isentropic level by means of multilinear regression. The regression function $f$ (see Eq. 7) includes regressors for the seasonal cycle, Quasi-Biennial Oscillation (QBO), El Niño Southern Oscillation (ENSO) and a long-term linear trend.

### 3.4.1 Seasonal cycle

In order to illustrate the seasonal cycle of the PVG tropopause, Fig. 5 displays the 1980 to 2017 monthly mean climatologies of tropopause latitude and PV on three isentropic surfaces, 330 K, 350 K and 370 K, determined from the aforementioned reanalyses. Figure 5a–f shows the latitude $\phi^{TP}$ of the tropopause intersections; the corresponding potential vorticity $PV^{TP}$ is shown in Fig. 5g–l below. Figure 5 reveals a distinct seasonal cycle of the PVG tropopause with amplitudes ranging between $4°$ to $15°$ latitude and 0.2 PVU to 2 PVU. Particularly, the tropopause latitude and PV both reach higher absolute values in

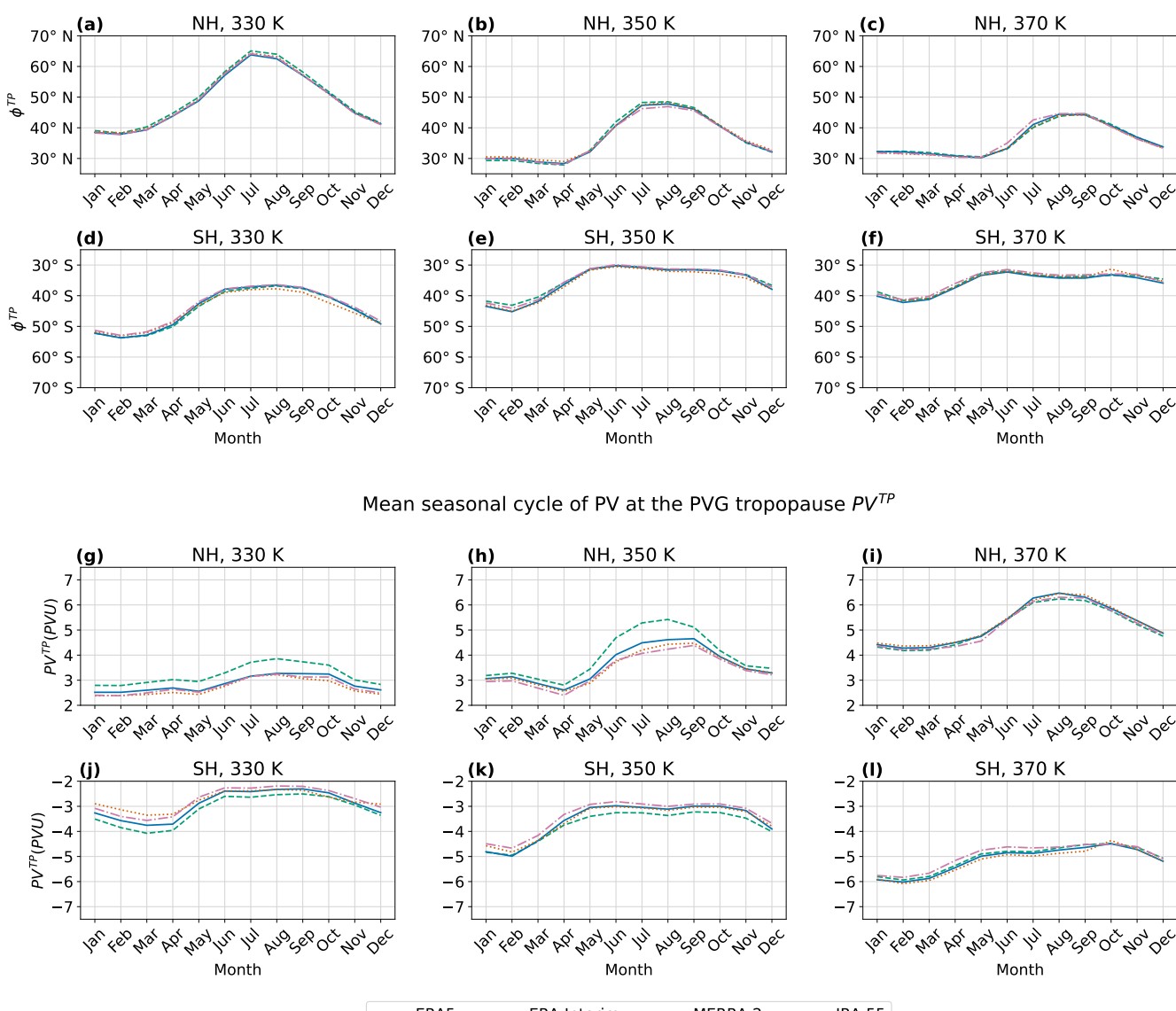

**Figure 5.** Mean seasonal cycle of the PVG tropopause averaged over equivalent latitude contours, showing latitude $\phi^{TP}$ (panels **a–f**) and potential vorticity $PV^{TP}$ (panels **g–l**) on different isentropic levels: $330\,\mathrm{K}$ (left column), $350\,\mathrm{K}$ (middle column) and $370\,\mathrm{K}$ (right column) in the northern hemisphere (panels **a–c** and **g–i**) and southern hemisphere (panels **d–f** and **j–l**). Curves display the monthly mean values of $\phi^{TP}$ and $PV^{TP}$ in the climatological period from 1980 to 2017 for each reanalysis ERA-Interim, ERA5, MERRA-2 and JRA-55.

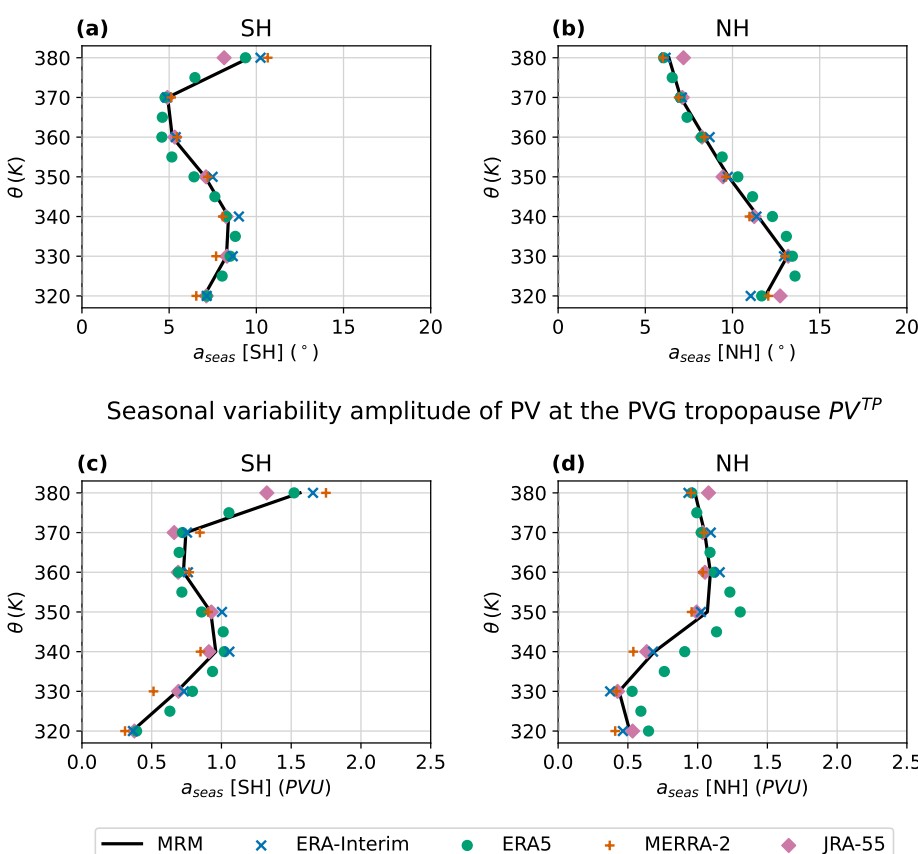

**Figure 6.** Seasonal variability amplitude $a_{seas}$ (see Eq. 7) of the PVG tropopause latitude $\phi^{TP}$ averaged over equivalent latitude contours (top row, panels **a** and **b**) and potential vorticity $PV^{TP}$ (bottom row, panels **c** and **d**) on isentropic levels between 320 K to 380 K in each hemisphere (northern hemisphere: left, southern hemisphere: right column), climatology of 1980 to 2017. Coloured symbols represent $a_2$ values for each reanalysis: in 5 K steps for ERA5 and 10 K for ERA-Interim, JRA-55 and MERRA-2. The arithmetic mean of these four reanalyses (multi-reanalysis mean, MRM) is drawn as a bold black line.

the summer of each hemisphere and lower absolute values in winter, which is consistent with the seasonal shift of the ITCZ. Comparing the three isentropes, Fig. 5a–c indicates a time lag of seasonality between the upper and lower isentropes in the NH, where the lower isentropes show an earlier onset of poleward tropopause movement than the upper levels. However, this time lag is not apparent in the SH in Fig. 5d–f. The seasonal variability of latitude and PV is qualitatively robust and the differences between different reanalyses do not exceed the seasonal variability. However, comparing Fig. 5a–f and Fig. 5g–l, the latitudinal seasonal cycles are more robust than those of potential vorticity, exhibiting smaller differences between reanalyses relative to

the annual amplitude. Notably from Fig. 5g–l, ERA5 yields substantially different $PV^{TP}$ values than the other three reanalyses, tending towards higher absolute PV on 330 K and 350 K.

To further examine the magnitude of seasonal variability in the PVG tropopause on different isentropic levels, Fig. 6 depicts vertical profiles of the amplitude $a_{seas}$ determined from multilinear regression (see Eq. 7). Therein, Fig. 6a–b displays seasonal variability amplitudes of the tropopause latitude $\phi^{TP}$, while the potential vorticity $PV^{TP}$ is featured in Fig. 6c–d below. The seasonal amplitudes are shown separately for the northern and southern hemispheres in the left and right column, respectively.

The seasonal cycle amplitude profiles in Fig. 6 exhibit differences between hemispheres and altitudes. Notably from Fig. 6a–b, the tropopause latitude varies substantially more in the northern hemisphere (Fig. 6b) than in the southern hemisphere (Fig. 6a), except for 380 K, where the amplitude is larger in the southern hemisphere. In the northern hemisphere (Fig. 6b), the seasonal amplitude reaches a maximum of around $13°$ latitude at 330 K and then continuously decreases with height until reaching a minimum of about $6°$ to $7°$ at 380 K. In the southern hemisphere (Fig. 6a), a local maximum of around $8°$ can be observed at 330 K to 340 K, a minimum of $5°$ at 370 K, and the maximum of around $9°$ at 380 K.

The seasonal PV variability in Fig. 6c–d tends to increase with height, and lies within the range of 0.25 PVU to 1.75 PVU. PV seasonality in the southern hemisphere (Fig. 6c) exhibits a similar vertical profile as the latitude, but in the northern hemisphere (Fig. 6d), PV variability is small compared to the latitudinal variability between 320 K to 350 K. During June–August, the PVG tropopause generally shifts northward, i. e., equatorward in the SH and poleward in the NH. In the climatological mean, this corresponds to a shift towards more positive PV at the tropopause. Compared to the large latitudinal shift, the PV variability in the northern hemisphere is smaller than expected. This mitigation of PV variability can be explained by the seasonal changes in the PV field itself, illustrated in Fig. B1. Contrasting the zonal mean climatologies of the PVG tropopause and PV fields in June–August against December–February, a decrease of PV is apparent in the mid-latitudes of the northern hemisphere between 320 K and 350 K. The subtropical jet is particularly weak in boreal summer, which can be seen in Fig. 1. Regarding Eq. 1 and Eq. 3, a weak subtropical jet also accounts for smaller PV. This June–August decrease of PV in the mid-latitudes between 320 K to 350 K mitigates the expected PV increase, which in turn decreases the seasonal PV variability on lower isentropes in the northern hemisphere as observed in Fig. 6d.

### 3.4.2 Interannual variability

This section further investigates the interannual variability of the PVG tropopause, comparing multi-linear regression amplitudes for the El Niño Southern Oscillation (ENSO) and Quasi-Biennial Oscillation (QBO).

Figure 7 shows the variability amplitude $a_{enso}$ related to El Niño Southern Oscillation (ENSO) of the PVG tropopause latitude $\phi^{TP}$ (Fig. 7a–c) and potential vorticity $PV^{TP}$ (Fig. 7d–e) as vertical profiles with respect to potential temperature. The left and middle columns display $a_{enso}$ in the southern and northern hemisphere, respectively. To further examine ENSO effects on the width of the tropical troposphere, Fig. 7c also shows the inter-hemispheric difference of tropopause latitudes, $\phi_{NH}^{TP} - \phi_{SH}^{TP}$, on each isentropic level. Overall, Fig. 7 shows that the tropopause variability attributed to ENSO is about one order of magnitude smaller than for the seasonal cycle, and further exhibits a vertical structure consistent between the different reanalyses.

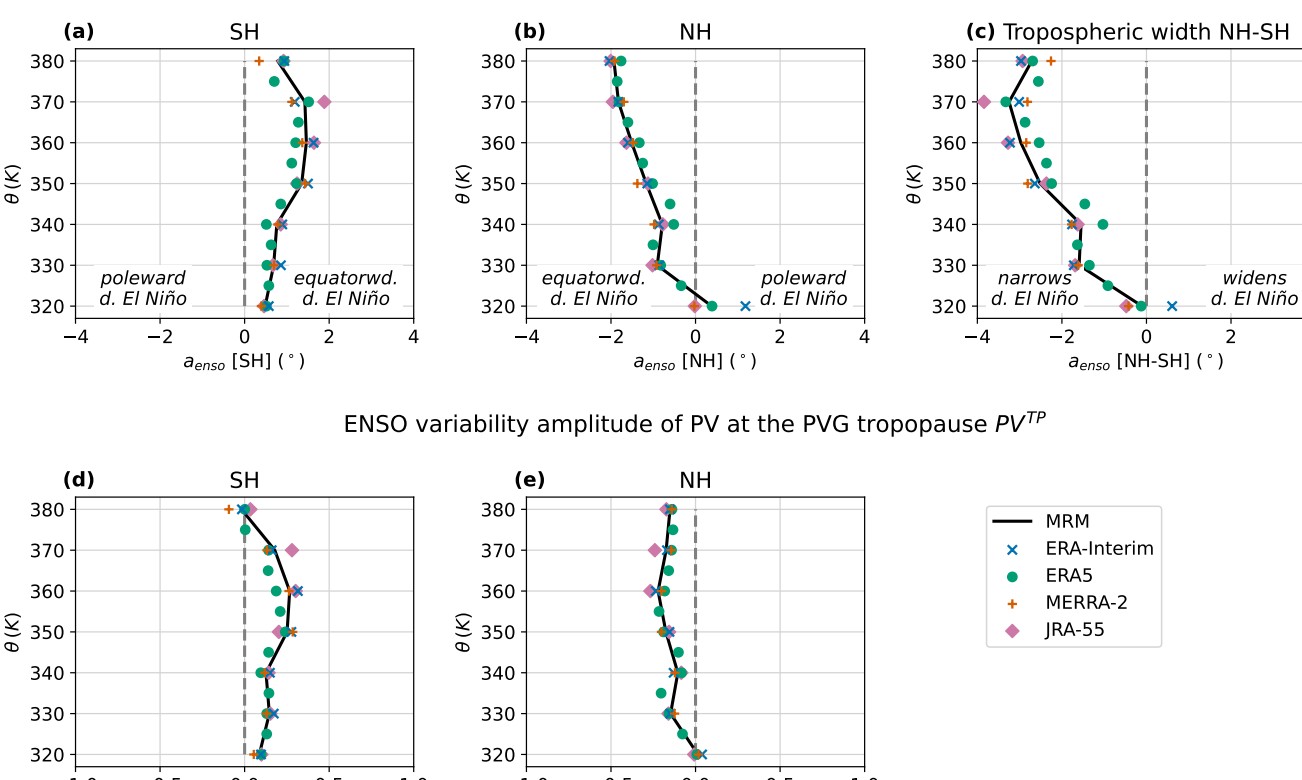

**Figure 7.** Vertical profile of the ENSO variability amplitude $a_{enso}$ in the PVG tropopause latitude $\phi^{TP}$ (averaged over equivalent latitude contours) and potential vorticity $PV^{TP}$; similar to Fig. 6. Additionally, the ENSO variability of the inter-hemispheric difference between tropopause latitudes, $\phi_{NH}^{TP} - \phi_{SH}^{TP}$, is shown in the rightmost panel **(c)**. The zero-crossing of $a_{enso}$ is marked by a dashed grey line. The annotations *poleward/equatorward d. El Niño* aim to indicate the direction in which the tropopause is shifted during El Niño; in the SH, positive amplitudes correspond to equatorward shifts and negative values to poleward shifts, and vice-versa in the NH.

In the southern hemisphere (Fig. 7a), the latitudinal variability amplitude for ENSO is positive on all levels and in all reanalyses, which relates to an equatorward shift of the PVG tropopause between $0.5°$ to $1.5°$ in El Niño years and a correspondent poleward shift during La Niña. This variability is most pronounced on the isentropic levels of $360\,\text{K}$ and $370\,\text{K}$, where the spread between the reanalyses is also largest. The variability in potential vorticity (Fig. 7d) agrees well with the variability in PVG tropopause latitude, exhibiting mostly positive amplitudes with a similar vertical structure.

In the northern hemisphere (Fig. 7b), the ENSO variability coefficients for the tropopause latitude are mostly negative, revealing equatorward displacements of the PVG tropopause in El Niño years and poleward shifts during La Niña—similar to the southern hemisphere. This variability tends to fortify with height, with amplitudes reaching up to $-2°$ latitude around

550

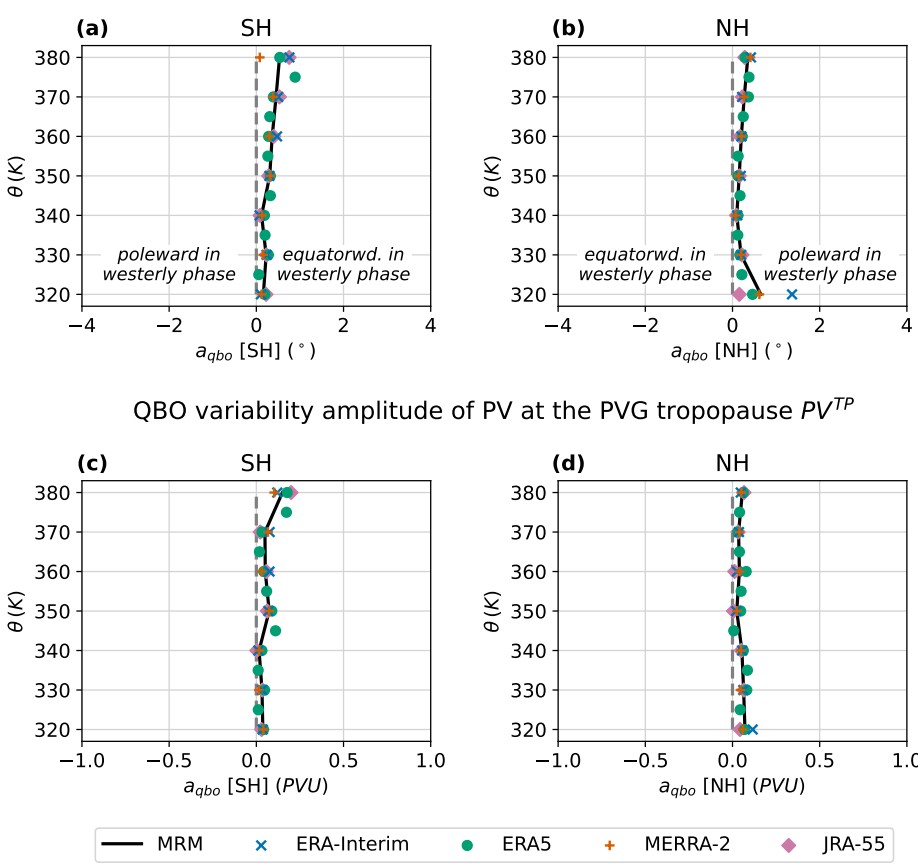

**Figure 8.** Vertical profile of the combined QBO variability amplitude $a_{qbo}$ of the PVG tropopause latitude and PV (averaged over equivalent latitude contours); similar to Fig. 6.

370 K–380 K. Overall, ENSO variability emerges similarly in the different reanalyses. The PV variability (Fig. 7e) exhibits a similar pattern with ENSO amplitudes close to zero at 320 K and larger negative values at upper levels. Above 360 K, the ENSO amplitude in tropopause PV variability slightly weakens. The negative variability amplitudes in tropopause PV above 320 K qualitatively match the equatorward tropopause shift during El Niño and poleward shift during La Niña.

Regarding upper tropospheric width as shown in Fig. 7c, a robust narrowing during El Niño and corresponding widening during La Niña can be observed at all isentropic levels and considered reanalyses, except at 320 K for MERRA-2 and ERA-Interim. The latitudinal ENSO variability reaches from $0°$ to $3°$, is weakest around 320 K and strongest around 370 K. Overall, the narrowing of the upper troposphere during El Niño and widening during La Niña, respectively, emerge robustly for all four reanalyses, which is qualitatively consistent with the contraction and expansion of the Hadley cells, as well as meridional shifts of the subtropical jets related to ENSO (Lu et al., 2008; Chen et al., 2008).

Figure 8 further shows the QBO variability amplitude $a_{qbo}$ of the PVG tropopause latitude. Here, the amplitudes for both QBO regressors, i.e., the tropical zonal wind speeds at $30\,\mathrm{hPa}$ and $50\,\mathrm{hPa}$, are combined (see Sect. 2.4). Overall, the QBO variability is one to two orders of magnitude smaller than the seasonal cycle, with amplitudes primarily confined between $0°$ to $1°$ latitude (Fig 8a–b) and corresponding $PV^{TP}$ up to $0.25\,\mathrm{PVU}$ (Fig 8c–d). Figure 8a shows positive QBO variability amplitudes $a_{qbo}$ in the southern hemisphere, indicating an equatorward shift of the tropopause with positive QBO index, i.e.,
during tropical westerly wind regimes, and poleward shifts during easterly QBO phases. In the northern hemisphere (Fig. 8b), $a_{qbo}$ is also positive on all levels, but represents a poleward shift during westerly phases, and an equatorward shift during easterly wind regimes—vice-versa to the southern hemisphere. Overall, QBO appears to shift the PVG tropopause northward during westerly phases and southward during easterly phases.

  In summary, the latitudinal location of the PVG tropopause in the northern and southern hemisphere shows a robust corre-
lation with ENSO. In particular, the upper troposphere narrows in El Niño years and broadens in La Niña years. The ENSO variability is negligible on the $320\,\mathrm{K}$ level, but increases with height, reaching $2°$ to $3°$ latitude on $370\,\mathrm{K}$–$380\,\mathrm{K}$. The PV variability qualitatively matches the latitudinal variability. The QBO effects are overall much weaker than the ENSO variability. QBO appears to affect the PVG tropopause more strongly on upper isentropic levels above the subtropical jet core, pushing the tropopause northward during westerly phases and southward during easterly regimes of tropical zonal winds.

### 3.4.3 Long-term trends

  Finally, the long-term linear trends in the PVG tropopause latitude (degrees per decade) and potential vorticity (PVU per decade) are shown in Fig. 9 as vertical profiles. The tropopause trends range between $\pm 0.5°$ latitude per decade and between $\pm 0.1\,\mathrm{PVU}$ per decade in each hemisphere. Hence, long-term trends are smaller than the seasonal cycle and ENSO-related variability, but larger than the QBO-related variability.

For the southern hemisphere, Fig. 9a exhibits noticeable vertical variation in the tropopause latitude trends, consisting of poleward shifts below $335\,\mathrm{K}$, equatorward shifts between $335\,\mathrm{K}$ and $375\,\mathrm{K}$, and zero to weak poleward shifts above in the multi-reanalysis mean. Throughout the profile, JRA-55 deviates most strongly from the other reanalyses, showing poleward shifts over the entire vertical range. However, the vertical shape of the JRA-55 trend profile is similar to the other reanalyses, with a local minimum at $350\,\mathrm{K}$. Potential vorticity as shown in Fig. 9d qualitatively reflects the latitude trends, with negative
PV changes below $335\,\mathrm{K}$, positive changes above, and JRA-55 largely deviating from the other reanalyses.

  In the northern hemisphere (Fig. 9b), latitudinal trends show an especially large spread between the reanalyses at $320\,\mathrm{K}$ and $380\,\mathrm{K}$. On the isentropic levels in between, the trend values are fairly robust, following a vertical structure similar to the southern hemisphere, with poleward shifts up to $345\,\mathrm{K}$, equatorward shifts between $350\,\mathrm{K}$ and $370\,\mathrm{K}$, and poleward shifts above. On the upper levels, JRA-55 again deviates from the other reanalyses, exhibiting almost no trend. Potential vorticity in
the northern hemisphere (Fig. 9e) follows a similar structure, indicating an increasing trend of PV up to $350\,\mathrm{K}$, decreasing PV up to $370\,\mathrm{K}$, and increasing PV above, which qualitatively matches the diagnosed latitude shifts.

  As apparent from Fig. 9c, the tropopause trends in each hemisphere cause a distinct vertical structure in upper tropospheric width trends. Below about $340\,\mathrm{K}$, the poleward latitudinal trends of the PVG tropopause in both hemispheres contribute to a

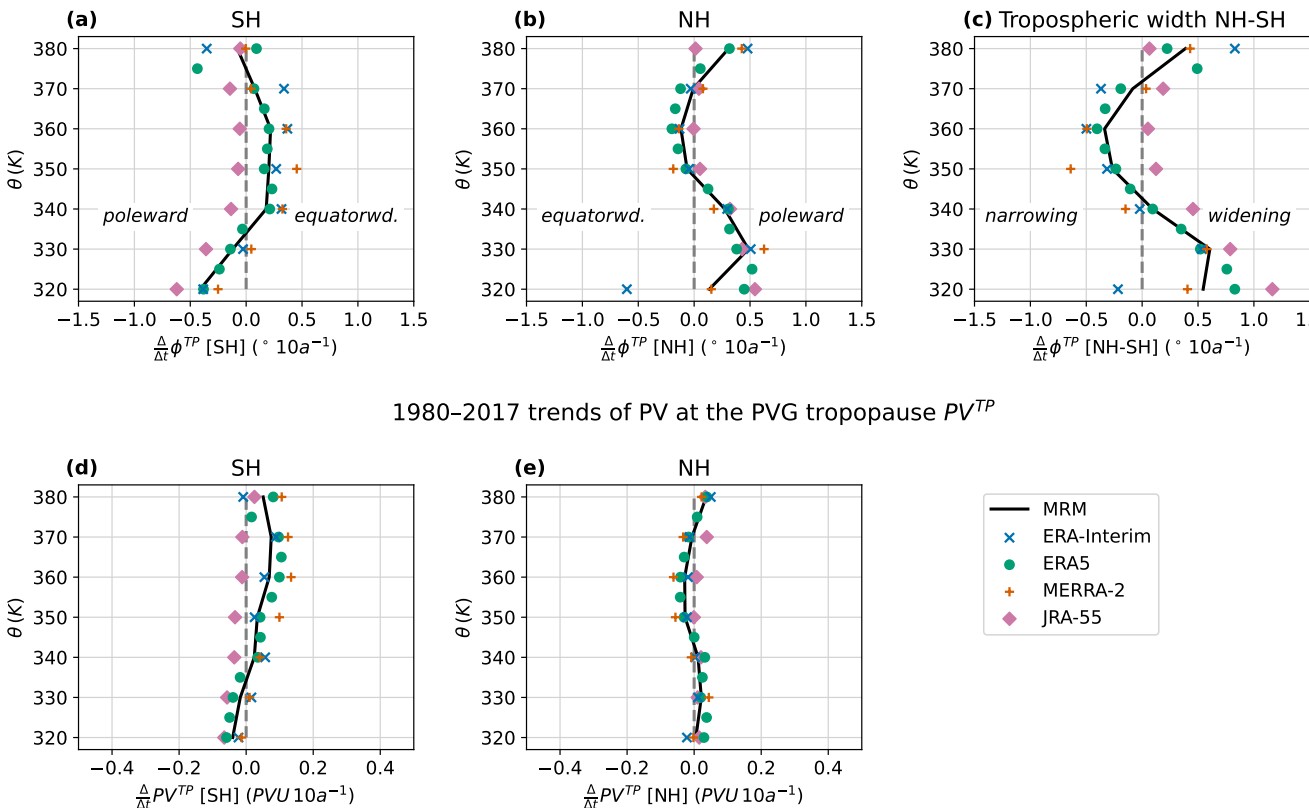

**Figure 9.** Linear long-term trends of the PVG tropopause (averaged over equivalent latitude contours) with respect to potential temperature ($\theta$) for four different reanalyses. Trends in tropopause latitude $\phi^{TP}$ are shown in panels **a–c**, the corresponding potential vorticity $PV^{TP}$ in panels **d** and **e**, for each hemisphere (left: southern hemisphere, middle: northern hemisphere, right: latitudinal difference between both hemispheres). Trends are determined as $a_{lin}$ from multilinear regression (see Eq. 7) and converted to units of degrees latitude per decade or PVU per decade, respectively.

poleward expansion of the troposphere of around $0.5°$ latitude per decade, which quantitatively matches tropical expansion rates as compiled by Staten et al. (2018, 2020) of $0.25°$ to $0.5°$ latitude per decade, and qualitatively corresponds to observed and modelled poleward shifts of the eddy-driven jet (Woollings et al., 2023). Above $340\,\mathrm{K}$ and up to about $370\,\mathrm{K}$, on the other hand, negative trends (equatorward shifts) in both hemispheres indicate a narrowing of the upper troposphere up to $-0.5°$ latitude per decade. This narrowing disagrees with aforementioned trends of tropical expansion (Staten et al., 2020), but matches other studies focusing on the tropopause. Martin et al. (2020), for instance, showed that the latitudinal distance of

the subtropical tropopause breaks in both hemispheres narrows by around $-0.5°$ per decade. Furthermore, Zou and Hoffmann (2023) determined a narrowing of the tropical troposphere of $(-0.16 \pm 0.11)°$ per decade. As the thermal tropopause gradient

metric developed by Davis and Rosenlof (2012) and employed by Zou and Hoffmann (2023) weighs higher altitudes more strongly, this narrowing trend possibly corresponds to the narrowing observed in the PVG tropopause between $340\,\mathrm{K}$ and $370\,\mathrm{K}$. The vertical structure in these changes of upper tropospheric width is qualitatively consistent for the different reanalyses, although JRA-55 numerically deviates from the other three reanalyses and yields no trends at the upper levels.

## 3.5 Simplifications of the PVG tropopause determination method

We examine potential simplifications of the PVG tropopause determination by comparing the original method (utilizing six-hourly reanalysis data, zonally resolved, including the STJ wind speed) to three alternative methods applied to the ERA5 reanalysis, which employ monthly climatologies of the input data, zonal means of the input data while computing gradients with respect to geographical latitude instead of equivalent latitude, and omit the STJ criterion, as detailed in Section 2.5. To evaluate how well these alternatives reproduce the PVG tropopause results from the original method, this section compares the seasonal multi-year climatologies, mean seasonal cycle, and trends computed with aforementioned methods.

Figure 10 shows that creating monthly averages of reanalysis data before applying the PVG TP algorithm does not substantially change the climatological latitude of the PVG TP between $340\,\mathrm{K}$ and $370\,\mathrm{K}$, but leads to some deviations on the lower and higher isentropes. Omitting the subtropical jet wind criterion introduces noise, as previously observed by Nash et al. (1996) in the polar vortex. Since the PV gradient can be enhanced in various regions—such as the edge of the Monsoon anticyclone (Ploeger et al., 2015)—the subtropical jet wind speed acts as an additional constraint helping to identify the maximum in the PV gradient that corresponds to the PVG tropopause. Averaging the reanalysis zonally before the PVG TP algorithm and simply computing the PV gradient with respect to common latitude instead of equivalent latitude alters the results drastically, as the PV-gradient method is originally intended to work with equivalent latitudes, i.e., on contours of PV, based on concepts by Butchart and Remsberg (1986), Nash et al. (1996) and Kunz et al. (2011). We therefore continue the following analyses only with the monthly mean, global reanalysis data including the wind criterion.

Figure 11 displays the climatological seasonal cycle of the PVG tropopause on three isentropic levels. On all isentropes, the seasonality of equatorward and poleward shifts is qualitatively similar in both datasets, but monthly climatologies tend to yield a more equatorward tropopause than the original six-hourly reanalysis data. This deviation is especially noticeable in the NH on $330\,\mathrm{K}$ (Fig. 11a) and the $370\,\mathrm{K}$ isentrope in boreal autumn (Fig. 11e), where the tropopause computed from monthly means is located $5°$ to $10°$ equatorward of the six-hourly tropopause product.

The long-term trends of the PVG tropopause computed from monthly climatologies are presented in Fig. 12. The variation with altitude is qualitatively similar for monthly climatologies and sub-daily data, indicating widening from $320\,\mathrm{K}$ to $340\,\mathrm{K}$, narrowing from $340\,\mathrm{K}$ to $370\,\mathrm{K}$, and widening above. However, using monthly climatologies yields smaller widening trends from $320\,\mathrm{K}$ to $340\,\mathrm{K}$, but larger narrowing trends from $340\,\mathrm{K}$ to $370\,\mathrm{K}$ than than those obtained from sub-daily data.

As a conclusion, we do not recommend using zonal mean reanalyses, since computing the PV gradient with respect to geographical latitude instead of equivalent latitude fundamentally changes the method. In addition, the STJ wind criterion in Eq. 5 is necessary, because the redundancy of the horizontal wind and PV gradient maxima at the tropopause break stabilizes the PVG tropopause calculation.

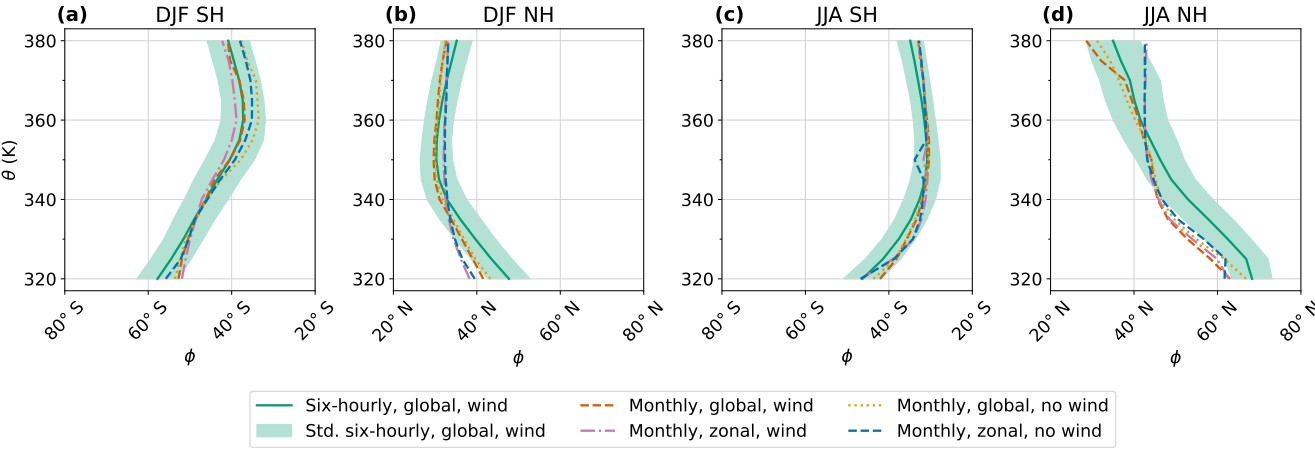

**Figure 10.** Comparison of methods for determining the PVG tropopause latitude $\phi^{TP}$ (averaged over equivalent latitude contours) from the ERA5 reanalysis, as detailed in Section 2.5. Seasonal climatologies (1980–2017) of $\phi^{TP}$ in each hemisphere are shown for December–February (DJF) in the southern hemisphere **(a)** and northern hemisphere **(b)**; for June–August (JJA) in the southern hemisphere **(c)** and northern hemisphere **(d)**. Lines indicate the mean climatology for each method; the standard deviation of the original method is shaded in green.

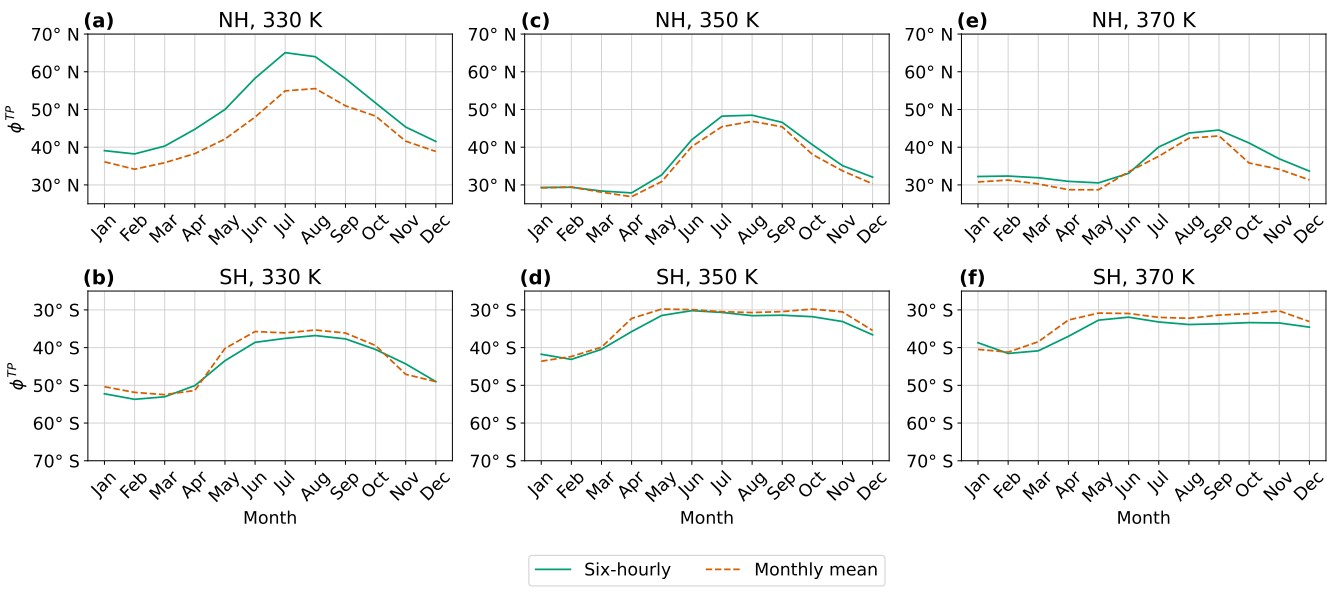

**Figure 11.** Climatological seasonal cycle (1980–2017) of the PVG tropopause latitude (averaged over equivalent latitude contours) in both hemispheres on three different isentropes. Similar to Fig. 5, but for ERA5 only; comparing the results from using the original six-hourly data versus monthly means as input.

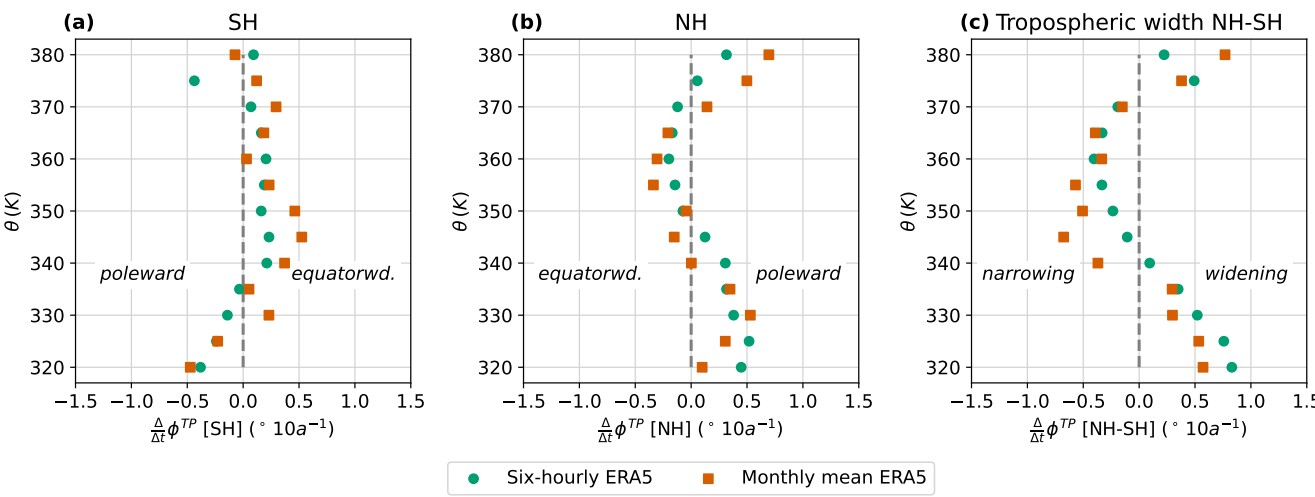

**Figure 12.** Linear long-term trends (1980–2017) of PVG tropopause latitude (averaged over equivalent latitude contours) with respect to potential temperature ($\theta$). Similar to Fig. 9, but for ERA5 only, comparing the results from the original six-hourly and monthly mean reanalysis data.

Computing the PVG tropopause from monthly averaged data yields long-term climatologies similar to the original six-hourly product in the range of $340\,\mathrm{K}$ to $370\,\mathrm{K}$, but the results may deviate noticeably on lower and higher isentropes. In addition, using monthly mean data tends to result in a more equatorward tropopause. Comparing long-term trends, the monthly means qualitatively match the six-hourly data, but result in much stronger (almost double) narrowing trends. Overall, using monthly averages efficiently reduces computational effort and yields similar climatologies in a certain range, but may not be sufficiently accurate in describing variabilities and trends. Based on these results, we advise using sub-daily to daily data for variability and trend analysis of the PVG tropopause.

### 3.6 Regional aspects

Given the significant regional variations in long-term trends observed for tropical width, the tropopause break (Martin et al., 2020), and the subtropical jets (Manney and Hegglin, 2018), this study conducts a similar regional analysis of the PVG tropopause break. The global tropopause surface, $\theta^{TP}(\phi, \lambda)$, was determined by identifying the corresponding PV contour on each isentropic level as detailed in Section 2.6. The results are presented in Fig. 13 as climatologies for the solstice seasons from 1980 to 2017, while the equinoctial seasons are shown in Fig. C1. These global fields reveal noticeable zonal variations in tropopause height and latitude, which can be attributed to undulations of the subtropical jet streams caused by Rossby waves, as well as baroclinic instability forming high- and low-pressure areas, leading to zonal variability in the PV field. The tropopause break is predominantly located between $30°$ and $40°$ latitude and is notably steeper in each winter hemisphere—visible in the

northern hemisphere in Fig. 13a and in the southern hemisphere in Fig. 13b. As the PVG tropopause is originally defined in the subtropics, the tropopause was extended poleward with the PV isosurface corresponding to the $320\,\mathrm{K}$ contour, which generally falls between $\pm 1\,\mathrm{PVU}$ to $\pm 4\,\mathrm{PVU}$, aligning well with traditional definitions of the dynamical tropopause (e. g., Hoerling et al., 1991; Holton et al., 1995).


Regional long-term trends of PVG tropopause latitude were computed at the tropopause intersections with each isentrope and every tenth meridian. As an example for the subtropical tropopause, the trends on the $330\,\mathrm{K}$ and $360\,\mathrm{K}$ isentropes in both hemispheres are presented in Fig. 14, resolved by longitude. On the $330\,\mathrm{K}$ isentrope, trends range between $0°$ and $0.75°$ per decade in the northern hemisphere and $-0.25°$ to $0.1°$ per decade in the southern hemisphere, indicating an overall tropical widening at lower levels. On the $360\,\mathrm{K}$ isentrope, trends vary between $-0.5°$ and $0.25°$ per decade in the northern hemisphere and $0°$ to $0.8°$ per decade in the southern hemisphere, which corresponds to an overall equatorward narrowing of the upper troposphere. The zonal structure of the tropopause trends at $360\,\mathrm{K}$ shows the strongest equatorward narrowing trends (negative in the NH, positive in the SH) at longitudes above the Pacific between $90°\mathrm{W}$ to $180°\mathrm{W}$. Comparing these PVG tropopause trends on the $360\,\mathrm{K}$ isentrope with the results of Martin et al. (2020) concerning trends in the tropopause break shows a similar longitudinal pattern: the strongest narrowing trends are found over the east Pacific in both studies.



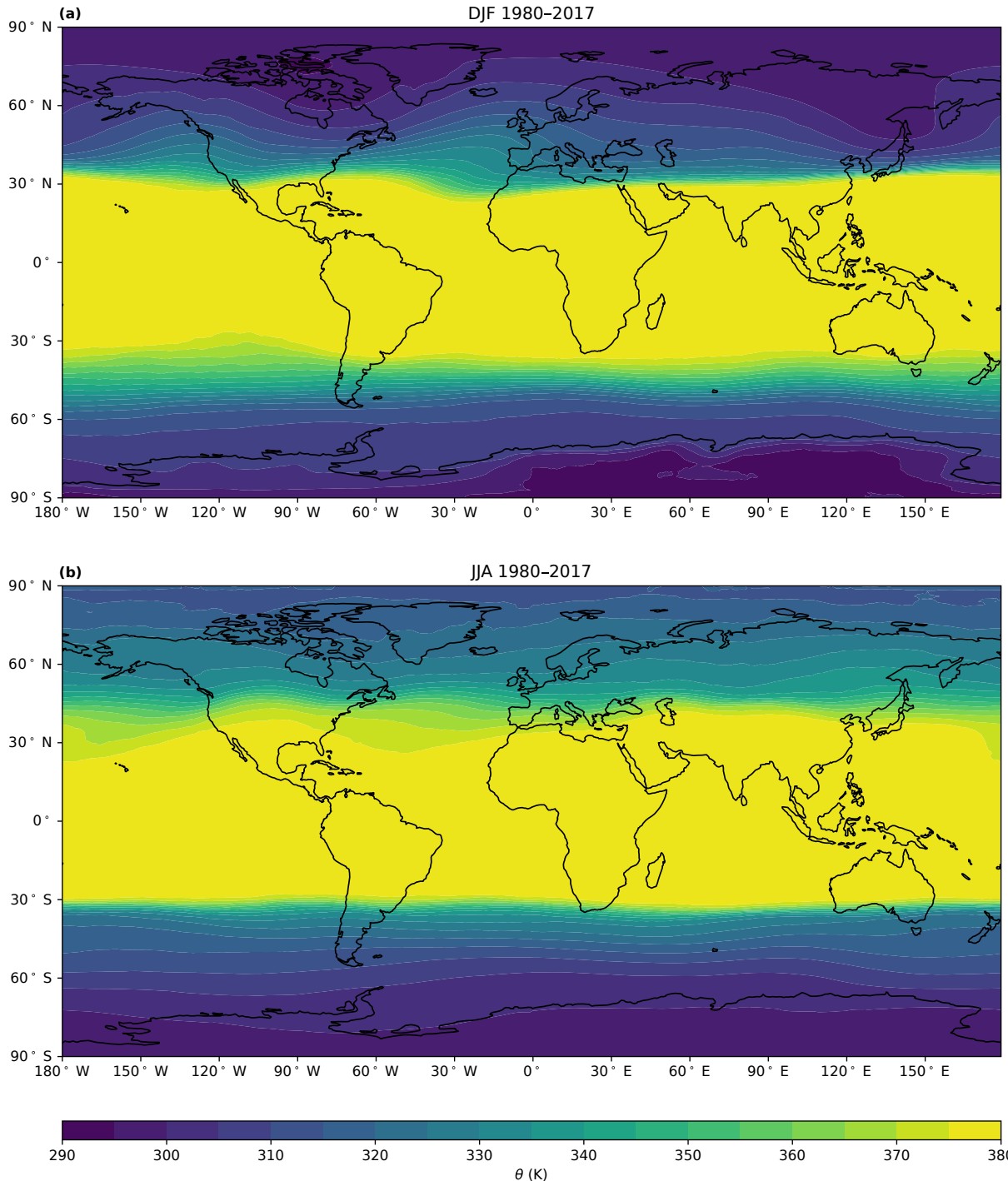

**Figure 13.** Global fields of PVG tropopause height in potential temperature coordinates $\theta^{TP}(\phi, \lambda)$ as a 1980–2017 climatology of the solstice seasons December, January, February (DJF) and June, July, August (JJA). Similar plots for the equinox seasons can be found in Fig. C1.

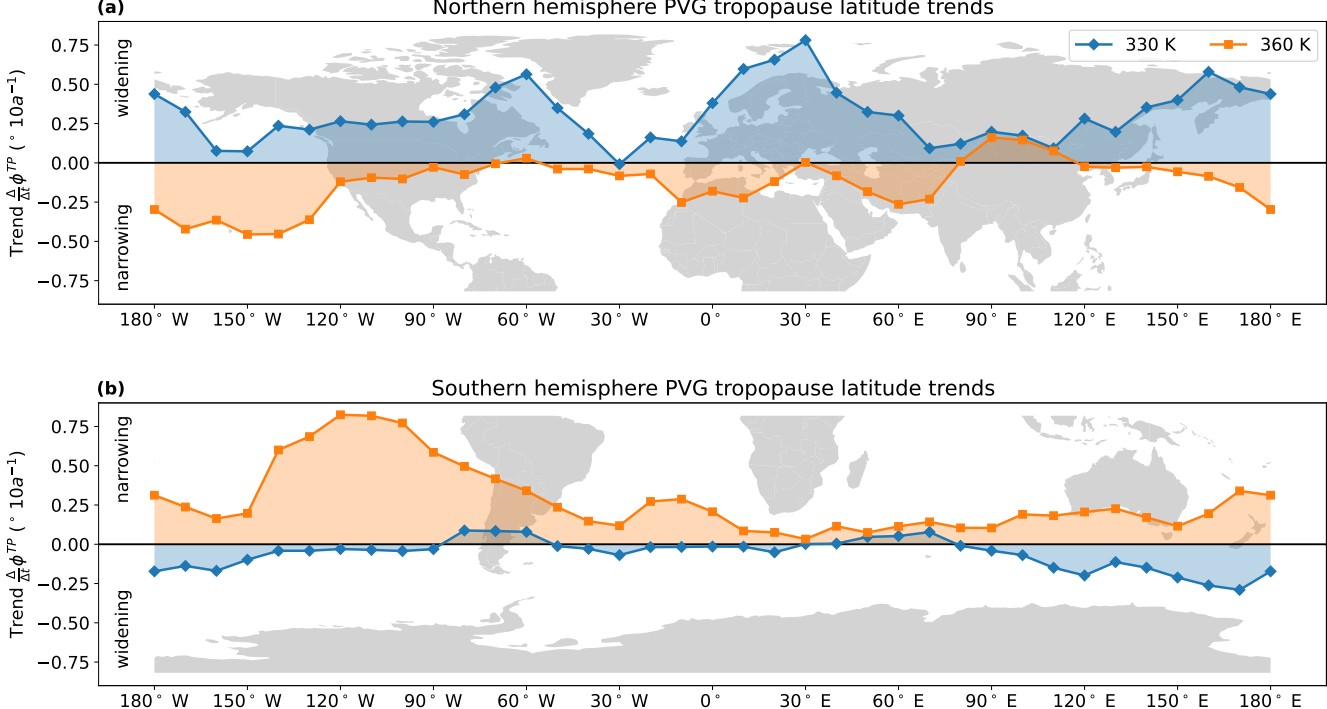

**Figure 14.** Zonally resolved long-term trends (1980–2017) of the PVG tropopause latitude on two isentropic surfaces, 330 K and 360 K, computed at every tenth meridian in the northern and southern hemisphere. As in Fig. 9, trends are displayed in degrees latitude per decade. In the northern hemisphere, positive trends indicate poleward widening and negative trends an equatorward narrowing. Conversely, in the southern hemisphere, positive trends signify equatorward narrowing, while negative trends indicate poleward widening.

## 4 Conclusions

The PV-gradient dynamical tropopause (PVG tropopause) has been introduced by Kunz et al. (2011) as a combination of meridional potential vorticity (PV) gradients and the subtropical jets (STJ) on each isentropic level, encompassing the mid-latitude to tropical tropopause. PV has been proven a useful variable in studying atmospheric circulation, since the conservation
of PV in adiabatic frictionless flow leads to the fact that PV is often distributed similarly to other species, for example trace gases (e. g., Hoskins et al., 1985; Hoskins, 1991). Additionally, strong meridional PV gradients in the subtropics, as well as the subtropical jet streams, act as barriers to quasi-isentropic meridional exchange between the upper troposphere and lower stratosphere (UTLS) in the subtropics (e. g., Holton et al., 1995). Considering these dynamical properties of PV and the jet streams, the PVG tropopause has the potential to reflect characteristics of transport in the UTLS more accurately than
conventional tropopause definitions.

This study compares the location of the PVG tropopause to other definitions, i. e., the thermal WMO tropopause and the "traditional" dynamical tropopause, which consists of a single PV isosurface. Since the PVG tropopause is directly linked

to UTLS transport barriers, changes in the PVG tropopause may indicate alterations of the global atmospheric circulation. Therefore, we furthermore examined the climatology, trends and variability of the PVG tropopause in the time range from 1980 to 2017 using four different reanalyses (ERA-Interim, ERA5, MERRA-2 and JRA-55) with six-hourly temporal resolution.

Our results show that the climatological location of the PVG tropopause computed from six-hourly reanalysis data is robustly represented in the four considered reanalyses and agrees well with the WMO lapse-rate tropopause in the subtropics and midlatitudes (Figs. 1, 3). The PV values at the PVG and WMO tropopause definitions vary strongly with season and altitude, ranging between $\pm 2\,\mathrm{PVU}$ at $320\,\mathrm{K}$ and $\pm 8\,\mathrm{PVU}$ at $380\,\mathrm{K}$ (Fig. 2). In accordance with Kunz et al. (2011), this large variability of PV shows that the tropopause is not well represented by any single PV isosurface. Therefore, the PV-gradient method is potentially more accurate in describing the dynamical tropopause than the traditional definition consisting of PV isosurfaces, e. g., the most commonly used $2\,\mathrm{PVU}$ surface.

Examining the global isentropic PV fields (Fig. 4) in all reanalyses, we find that the latitude of the maximum PV gradient is consistent in all reanalyses, but PV values at the latitude of the maximum PV gradient vary. Therefore, the latitude of the PVG tropopause is considerably more robust than the corresponding potential vorticity value. For following studies using the PVG tropopause, we therefore suggest to derive the PVG tropopause and other variables from the same reanalysis, to avoid errors stemming from differences in PV fields.

Potential simplifications of the PVG tropopause determination method were assessed, including using monthly climatologies instead of sub-daily data, zonal mean climatologies, and omitting the STJ wind criterion. Comparison with the standard method shows that using monthly means yields sufficiently accurate results near the tropopause break between $340\,\mathrm{K}$ and $370\,\mathrm{K}$ if the long-term climatology is considered, but trends computed from monthly means differ noticeably from the sub-daily product. An attempt to determine the PVG tropopause from zonal mean data showed considerable deviations from the original result, since the method is designed for zonally resolved data. Omitting the STJ wind criterion leads to increased noise and fluctuations in the PVG tropopause latitude. Therefore, for climatological analysis the PVG tropopause can be calculated from monthly mean, zonally resolved data if the STJ criterion is included. For trend and variability analysis we strongly recommend computing the PVG tropopause on a (sub-)daily timescale and taking averages thereafter.

A multilinear regression analysis of the PVG tropopause time series between 1980 and 2017 reveals that the modes of variability are consistent in the four considered reanalysis, including the seasonal cycle, ENSO and QBO, as well as long-term trends, which are reflected both in the zonal mean latitudinal position as well as PV of the tropopause. The seasonal cycle accounts for most of the variability, shifting the tropopause north- and southward by $5°$ to $15°$ latitude, which is concurrent with the shift of the ITCZ (Figs. 5, 6).

The PVG tropopause varies substantially with the El Niño Southern Oscillation (ENSO); associated latitudinal shifts range from $0°$ to $4°$ latitude and increase with altitude (Fig. 7). We found the tropical tropopause to narrow during El Niño and widen during La Niña, a result which is consistent with the ENSO variability of the Hadley cells and subtropical jet latitudes (Lu et al., 2008; Chen et al., 2008). The variability of the PVG tropopause with the Quasi-Biennial Oscillation (QBO) is qualitatively robust, but less pronounced than ENSO: during westerly phases of equatorial zonal winds, the PVG tropopause appears to shift northward in both hemispheres, and southward in easterly phases of QBO (Fig. 8).

The long-term trends of the PVG tropopause mostly range between $\pm 0.5°$ latitude per decade and exhibit a distinct vertical structure. Except for JRA-55, the tropopause shifts poleward between $320\,\text{K}$ and $340\,\text{K}$, narrows equatorward between $340\,\text{K}$ and $370\,\text{K}$, and expands poleward between $370\,\text{K}$ and $380\,\text{K}$ in both hemispheres (Fig. 9). Comparing our results to published trends of tropical width and the jet streams, this vertical structure has not been explicitly resolved before. Several studies suggest a widening of the tropics around $0.25°$ to $0.5°$ latitude (Staten et al., 2020) as well as poleward shifts of the subtropical and eddy-driven jets (Archer and Caldeira, 2008; Woollings et al., 2023). However, other studies focusing on tropical tropopause width showed narrowing trends in some regions and time ranges, which result in an overall narrowing of the tropical troposphere (Martin et al., 2020; Zou and Hoffmann, 2023).

In order to examine regional variations, we computed global fields and zonally resolved trends of the PVG tropopause. The regional trends confirm our findings of tropical widening at lower levels and upper tropospheric narrowing in the zonal mean view. Strong latitudinal variability is apparent in the trends; notably, trends on the $360\,\text{K}$ isentrope exhibit a latitudinal pattern very similar to that observed by Martin et al. (2020) in the tropical WMO tropopause break, with the strongest narrowing occurring over the East Pacific.

We hypothesize that the poleward trends of the PVG tropopause on lower isentropes could be related to poleward trends of the eddy-driven and subtropical jets found by Woollings et al. (2023). Specifically, the eddy-driven jets have been found to be strongly correlated with the latitudes of subtropical downwelling at the poleward edges of the Hadley cells, and therefore to tropical width (Davis and Birner, 2017; Menzel et al., 2019). Since eddy-driven and subtropical jets occasionally coalesce, the poleward expansion of the PVG tropopause around $320\,\text{K}$ is likely related to the poleward trends of the eddy-driven jets, and associated poleward expansion of the Hadley cells. The equatorward shifts of the PVG tropopause between $340\,\text{K}$ and $370\,\text{K}$ match narrowing trends in the tropical tropopause (Martin et al., 2020; Zou and Hoffmann, 2023). It needs to be noted that the currently available 40-year span of reanalysis data might be too short to discern long-term trends from natural variability (e. g., Woollings et al., 2023). Subsequent analyses are needed to understand this vertical structure of PVG tropopause trends, and possible relation to different aspects of the global atmospheric circulation.

## Appendix A: Climatological structure of the PVG tropopause in equinox seasons

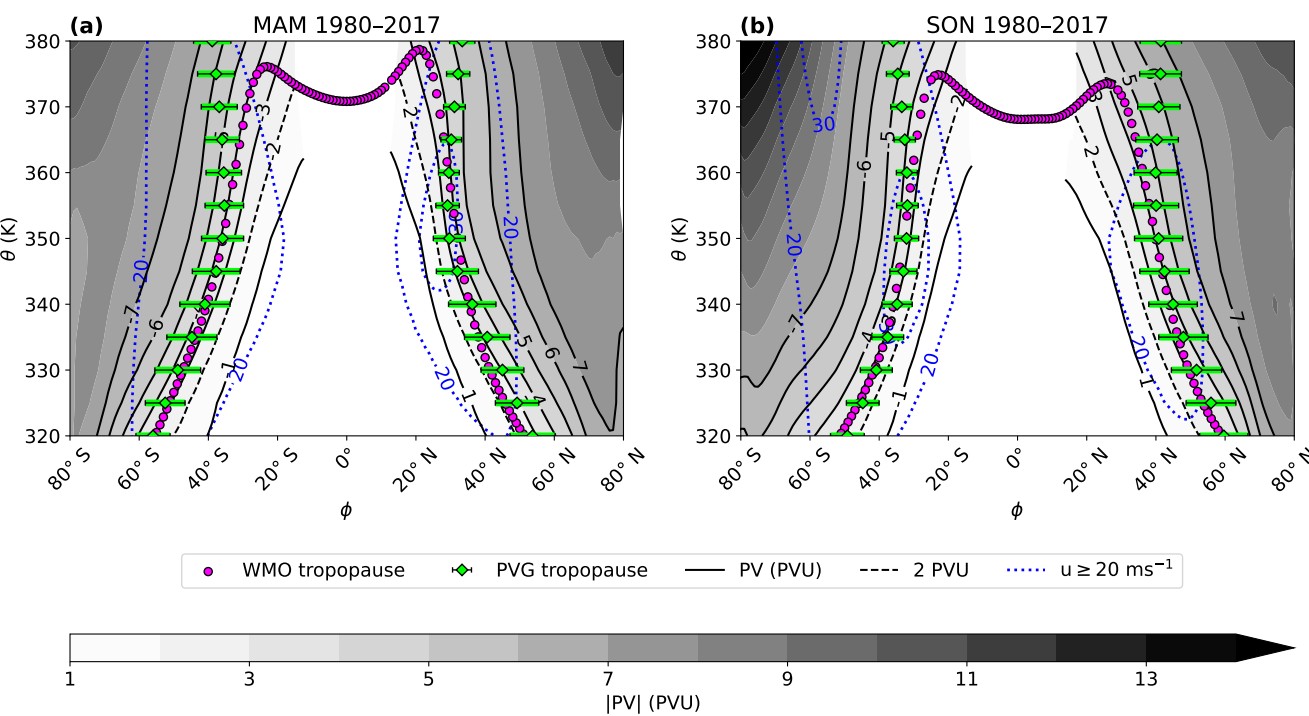

**Figure A1.** Seasonal climatology in **(a)** March–May (MAM) and **(b)** September–November (SON) 1980–2017 of the PVG tropopause latitude $\phi^{TP}$ averaged over equivalent latitude contours, compared to the WMO lapse-rate tropopause in front of the PV-field (grayscales with solid black PV isolines between $\pm 1$ and $\pm 7$ PVU) and the zonal wind $u$ maxima indicating the locating of the subtropical jets (dotted blue contours), displayed in the latitude/potential temperature plane. A similar figure has been published by Kunz et al. (2011) for ERA-Interim and is recreated here in a slightly altered form for ERA5.

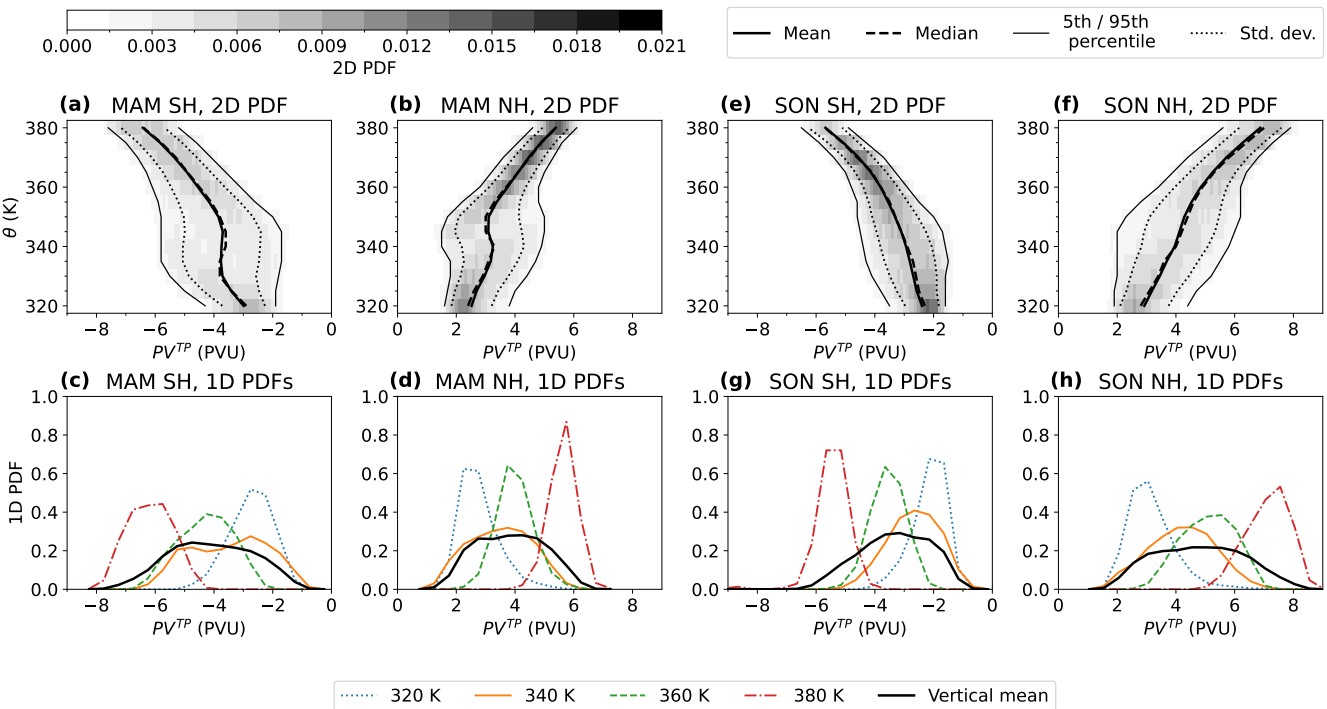

**Figure A2.** Seasonal climatology of the PV distribution at the PVG tropopause $PV^{TP}$ as probability density functions (PDFs) for ERA5 between 1980 and 2017, following Kunz et al. (2011). The left-hand side (panels **a–d**) shows the March–May (MAM) climatology, while the right-hand side (panels **e–h**) presents September–November (SON). The upper panels (**a, b, e, f**) display the 2D PDF in the PV, $\theta$ plane to illustrate the change of the PV distribution with height, with a bin size of 0.1 PVU and 5 K for ERA5, 10 K for all other reanalyses. Additionally, the mean, median, $1\sigma$ interval, 5th and 95th percentile of the PDFs are delineated. In the lower panels (**c, d, g, h**), the corresponding 1D PDFs on selected isentropes are drawn as coloured lines. The vertical mean 1D PDF along the whole tropopause across 320 K to 380 K is specified as a heavy black line.

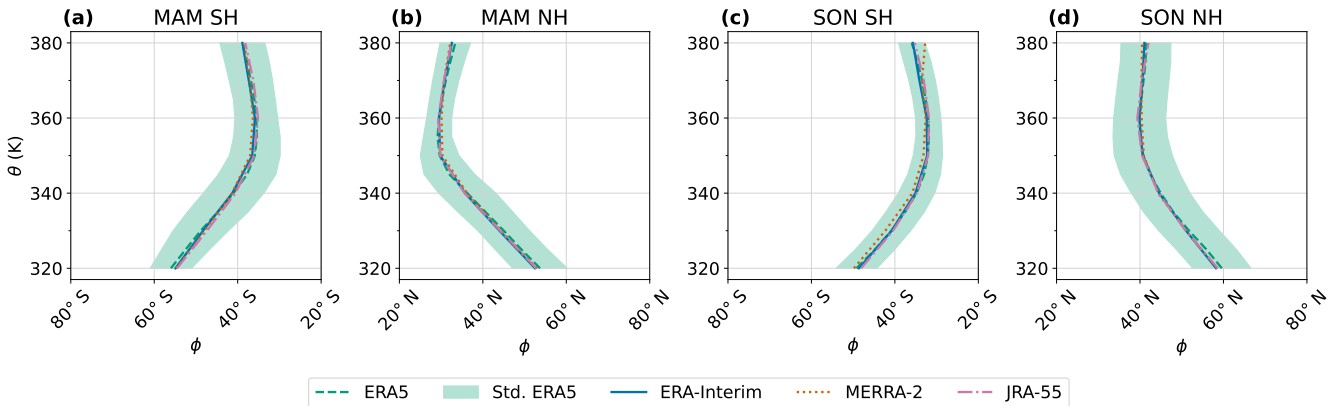

**Figure A3.** Comparison of the PVG tropopause latitude $\phi^{TP}$ averaged over equivalent latitude contours in four reanalyses. Seasonal climatologies (1980–2017) of $\phi^{TP}$ in each hemisphere are shown for March–May (MAM) in the southern hemisphere **(a)** and northern hemisphere **(b)**; for September–November (SON) in the southern hemisphere **(c)** and northern hemisphere **(d)**. Lines indicate the mean climatology in each reanalysis, the standard deviation of ERA5 is shaded in green.

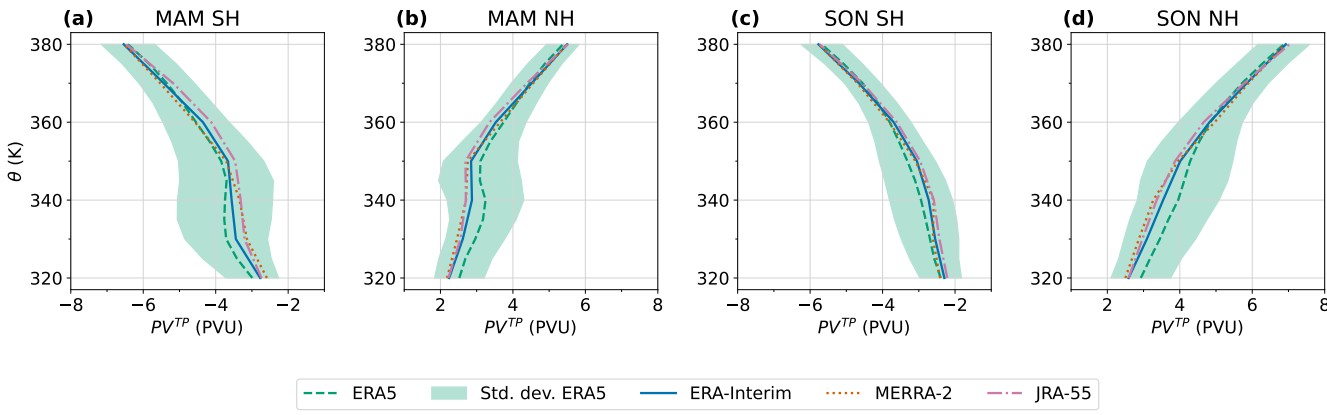

**Figure A4.** Means and standard deviations of the PDFs of $PV^{TP}$ (as shown in the upper panels of Fig. A2) in four reanalyses. Similarly to Fig. 3, the seasonal climatologies between 1980 and 2017 are shown for each hemisphere in March–May (MAM) in panels **(a)** and **(b)**, for September–November (SON) in panels **(c)** and **(d)**. Lines indicate the $PV^{TP}$ climatological means in each reanalyses; the standard deviation of ERA5 is shaded in green.

## Appendix B: Seasonal variability

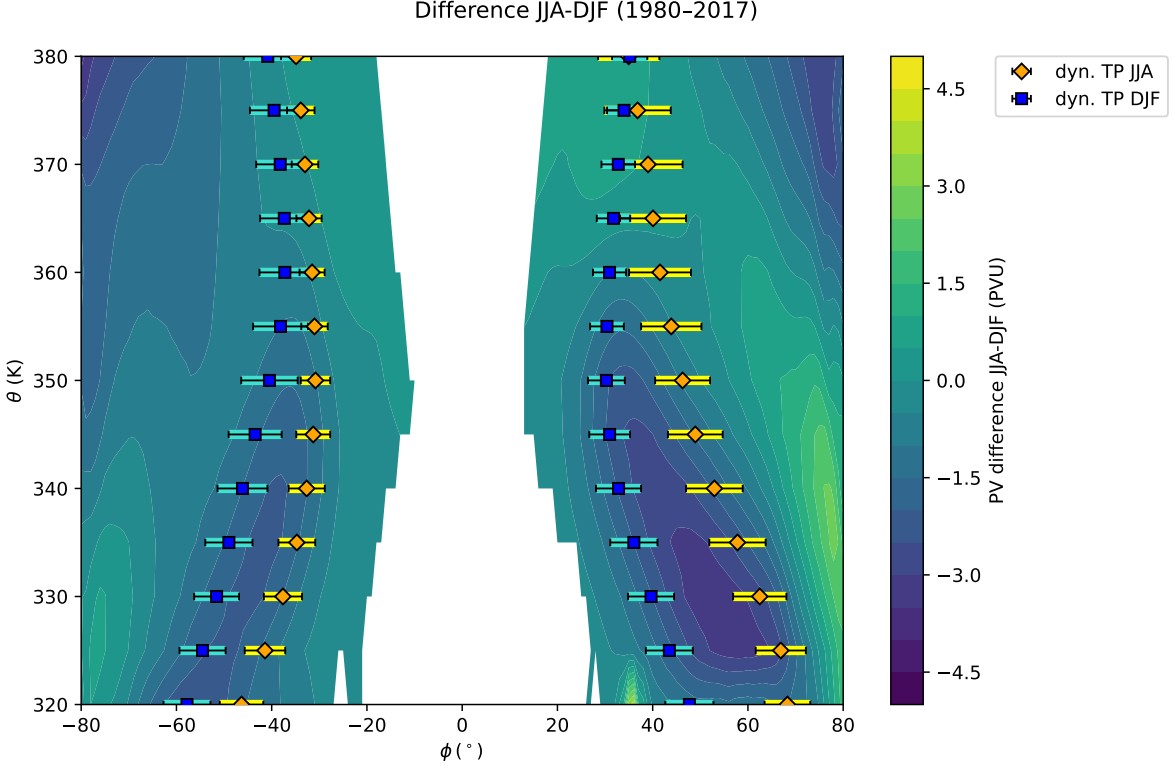

**Figure B1.** Difference of the potential vorticity fields and PV-gradient (PVG) dynamical tropopause, seasonal climatologies JJA-DJF in 1980 to 2017.

## Appendix C: PVG tropopause global fields in equinox seasons

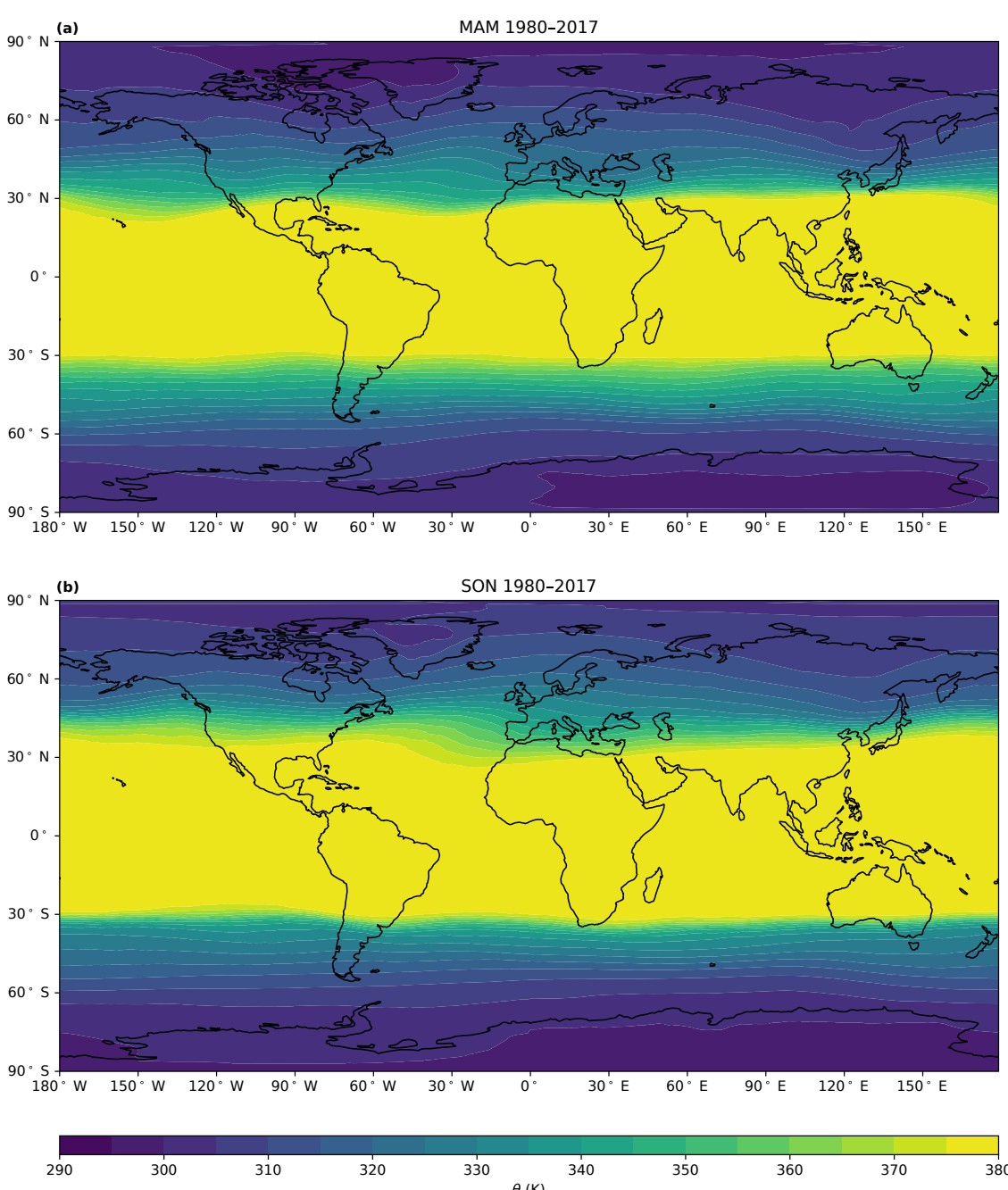

**Figure C1.** Global fields of PVG tropopause height in potential temperature coordinates $\theta^{TP}(\phi, \lambda)$ as a 1980–2017 climatology of the equinox seasons March, April, May (MAM) and September, October, November (SON). Similar plots for the solstice seasons can be found in Fig. 13.

*Data availability.* Time series of the PVG tropopause from 1980 to 2017 in the reanalyses ERA5, ERA-Interim, MERRA-2 and JRA-55 have been published alongside this paper by Turhal (2024). ERA5 and ERA-Interim reanalysis data have been provided by the European Centre for Medium-range Weather Forecasts (ECMWF). The complete ERA5 dataset is available via Hersbach et al. (2017) and the low temperature bias correction ERA5.1 from Simmons et al. (2020a). For ERA-Interim, the ECMWF Public Datasets Service closed on 1 June 2023; however, the dataset can still be downloaded programmatically from Dee et al. (2011). The JRA-55 reanalysis data is available at Japan Meteorological Agency, Japan (2013). The MERRA-2 dataset is provided by Global Modeling and Assimilation Office (GMAO) (2015) of the National Aeronautics and Space Administration (NASA). The Multivariate ENSO Index Version 2 (MEI.v2) is available from the National Oceanic and Atmospheric Administration (NOAA) Physical Sciences Laboratory; QBO time series have been compiled by Freie Universität Berlin.

*Author contributions.* FP, TB, PH and PK conceptualized the core research questions and goals; JC and KT developed and maintained the software; KT performed investigation, data analysis, data curation and visual representation; FW validated the results; KT and FP wrote the manuscript draft; FW, JC, PH, PK and TB reviewed the manuscript.

*Competing interests.* The authors declare that they have no conflict of interest.

*Acknowledgements.* This research was funded by the Deutsche Forschungsgemeinschaft (DFG, German Research Foundation) under the project titled "TPChange, The Tropopause Region in a Changing Atmosphere", designated as DFG TRR 301 with the Project-ID 428312742. Our gratitude extends to both TPChange and the Institute of Climate and Energy Systems, Stratosphere (ICE-4) at the Research Centre Jülich, for creating an inspiring and supportive community in atmospheric science. Special thanks are due to Frederik Harzer of Ludwig-Maximilian University of Munich for insightful discussions. Acknowledgment is also due to the Jülich Supercomputing Centre (JSC) at the Research Centre Jülich for allocating computational resources for our data analysis as part of the VSR project with the ID CLAMS–ESM. We extend our thanks to Nicole Thomas, Verena Alishahi, and Reimar Bauer (ICE-4), along with the JSC Support team at Research Centre Jülich, for their computational guidance and technical assistance. We gratefully acknowledge the reanalysis data provided by ECMWF, JMA, and NASA, and express our gratitude to Jens-Uwe Grooss and Lars Hoffmann (ICE-4, Research Centre Jülich) for their efforts in processing and making available the reanalysis data on the Research Centre's computational infrastructure. Last but not least, we greatly appreciate the careful reading of the manuscript and helpful feedback by the editor Laura Wilcox, two anonymous reviewers, as well as Tim Blazytko, Jan Conrads and Y. C..

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
