# Peer review of "Variability and trends in the PV-gradient dynamical tropopause"

_EGUsphere, 2024_

## Author Comment (AC1)

**Reply to Reviewer 1**

We thank the Reviewer for the careful reading and evaluation of the manuscript and the good comments which helped a lot to further improve the paper. In the following, we address all comments and questions raised (Reviewer's comments in italics).

We see two main concerns raised by the Reviewers, regarding (i) the complexity of the PVG tropopause determination algorithm and (ii) the analyses being carried out in zonal mean view only, which masks regional variations. A short overview of the related changes in the revised manuscript is:

**(i) Simplification of the algorithm**

We acknowledge the concern regarding the computational expense of determining the PVG tropopause from sub-daily global reanalysis data with several additional criteria concerning the height of maxima, latitude boundaries, and wind speed. In response, we have conducted additional analyses to examine the feasibility of simplified methods using monthly averaged data and the potential impact on the accuracy of the PVG tropopause latitude estimation. Specifically, we tested the following alterations:

1. Monthly reanalysis climatologies

2. Monthly reanalysis climatologies without the wind criterion

3. Monthly and zonal mean reanalysis climatologies

4. Monthly and zonal mean reanalysis climatologies without the wind criterion

These methods were applied to ERA5 reanalysis data, and we compared the results with the standard method that uses six-hourly, global reanalysis data with the wind criterion.
Our findings indicate that using monthly averages does not significantly alter the climatological latitude of the PVG tropopause between 340K and 370K, though deviations are observed at lower and higher isentropes. Omitting the wind criterion leads to increased noise, corroborating findings by Nash et al. (1996), due to the stabilizing redundancy of the PV gradient and horizontal wind peaks. Zonally averaging the reanalysis data and computing the PV gradient with respect to geographical latitude instead of equivalent latitude resulted in substantial changes to the results, as the method is designed to work with equivalent latitudes (cf. Nash et al. 1996, Kunz et al. 2011).
Based on these results, we do not recommend using zonal mean reanalyses or omitting the wind criterion. Averaging reanalyses over months yields similar results for the tropopause climatology near the tropopause break between 340K and 370K, but notable deviations above and below. The trends computed from monthly climatologies qualitatively show a similar vertical structure, but deviate numerically from the trends computed from sub-daily data. Therefore, using monthly climatologies may be sufficient if the focus is on the climatological structure of the tropopause break, but trends should be computed with sub-daily data.
The addidtional analyses of method simplifications are detailed in two new sections in the manuscript: Methods section 2.5 and Results section 3.5.

**(ii) Regional climatology and long-term trend analysis**

We agree that using a global zonal mean in our analysis could mask significant regional (longitudinal) differences. To address this, we have conducted a supplementary regional analysis focusing on these variations in the climatology and long-term trends of the PVG tropopause.
By identifying the PV contour for the tropopause in ERA5 data at each isentropic level, we determined global surfaces of the tropopause $\theta(\phi, \lambda)$. Poleward of the 320K contour, we continued the PVG tropopause by setting the tropopause as the PV isosurface corresponding to the PV value of the tropopause on 320K.
We computed zonally resolved long-term trends by applying multilinear regression to the tropopause intersection at each longitude and isentrope. The results confirm that the mid-latitude tropopause widens poleward overall in both hemispheres, while the tropical tropopause narrows equatorward. These trends vary noticeably with longitude. Comparison with Martin et al. (2020) reveals a similar longitudinal pattern; specifically, the strongest narrowing trends are observed over the east Pacific.
The corresponding methods and results are described in additional subsections of the Methods chapter 2.6 and Results 3.6.

A more detailed reply to all comments and description of the changes in the revised manuscript is given below.

**1 Overall comments**

*Tropical circulation and so-called "tropical width" changes are important phenomena to better understand in the context of climate variability and change. The paper "Variability and trends in the PV-gradient dynamical tropopause" by Turhal et al. examines a new definition for tropical width based on a potential vorticity-based definition that is applied to several reanalyses. This paper is very well written and thorough, and I have no doubt it will be of great interest to ACP readers. I have mostly minor suggestions to clarify a few points in the manuscript.*

Thank you for this positive evaluation of the manuscript!

**2 General comments**

*The only "major" point I would like to suggest is that the authors consider whether their rather more involved methodology (which involves computing PV, etc. from the reanalysis model level fields at high temporal frequency and complicated peak finding rules) is really necessary in order to accurately quantify the seasonal to multidecadal variability in the PVG tropopause latitudes? I have no objection to applying this more "expensive" approach to analyzing the data if it is truly necessary, but it would be helpful for future users to know whether it really is necessary, or whether more simplified approaches with averaged fields could be used without a loss of fidelity (e.g., monthly mean PV/wind fields on pre-determined isentropic levels, as is commonly provided by reanalysis centers)? The answer to this question has implications for how easily this type of analysis might be extended in time or to other reanalyses, as well as how straightforwardly it might be applied to model output fields from multi model experiments (where high temporal / vertical resolution fields are often not made available). If the authors could provide some insight on this issue, I think it would be a valuable contribution to the community.*

This is a very good point! Determining the PVG tropopause from sub-daily reanalysis data over 40 years is indeed computationally expensive. As explained above in our main point (i), we have therefore run additional tests with monthly climatologies of the ERA5 reanalysis and examined the influence of using zonal mean data and omitting the wind criterion, as detailed in a new section 2.5 in the Methods chapter. We computed the seasonal climatologies, mean seasonal cycle and long-term trends from these additional runs, which are presented in a new section "3.5 Simplifications of the PVG tropopause determination method" in the results chapter.

**3 Specific comments**

*Line 53: I suggest defining PVU at its first use (i.e., the SI unit equivalent)*

Thank you! Rephrased the paragraph at lines 44-48 as follows and included the unit definition of PVU:

"Another common definition of the tropopause, the so-called *dynamical tropopause*, is based on the *potential vorticity (PV)*, an analogue of angular momentum in air flow introduced by Rossby (1940) and Ertel (1942). PV is measured in *Potential Vorticity Units (PVU)* with $1\,\mathrm{PVU} = 10^{-6}\,\mathrm{m^2 s^{-1} K\, kg^{-1}}$. An invertibility principle holds which allows inferring the flow velocity field from the PV distribution. PV is therefore closely linked to atmospheric dynamics (Hoskins, 1985), which makes it particularly valuable for transport studies. "

*Line 84: Suggest citing Waugh et al 2018 here.*

Thank you for the suggestion! Replaced the other references with "Waugh et al, 2018" (now line 87).

*Line 97: Suggest citing Santer et al 2003 here*

Added the reference, thank you!

*Line 114: I think "contracted" might be a more appropriate word here than "converged"*

Thanks for pointing out this misleading wording! Changed it to "while the STJ of both hemispheres converged towards each other".

*Line 119: "to which extent" → "to what extent"*

Fixed!

*Line 140 − 145: This intro section confused me at first because it seemed to not be a detailed enough description, but then I eventually figured out that the material was discussed in further detail in the following subsections. You might consider referencing the subsections here to make it clear that further details are provided there.*

*Line 140 − 141 (and elsewhere): I found myself confused a few times in the manuscript regarding some of the uses of words like "gradient" where the direction of the gradient was not really specified. In this sentence (and throughout the paper) the gradients being referenced are meridional gradients on isentropes. Similarly, there's a bit of potential confusion around the discussion of the PVG dynamical tropopause being something that is defined based on a meridional gradient, versus what most people think of which is the tropopause being defined based on a vertical gradient/threshold. It might be worth considering the language in the paper and whether it would be more appropriate to use the term PVG dynamical tropopause \*width\* when referring to the latitude, rather than just calling it the PVG dynamical tropopause.*

Indeed! We reworked the intro section (now lines 143-151) and added references to the individual subchapters, and clarified that we compute a meridional gradient:

"The PV gradient-based (PVG) dynamical tropopause (Kunz, 2011) is determined as a contour on surfaces of equal potential temperature (i.e., *isentropes*) from the meridional gradient of potential vorticity (PV), combined with the location of the subtropical jet streams. This study is based on four different meteorological reanalyses: ERA-Interim, ERA5, MERRA-2 and JRA-55, which are described in Section 2.1. From six-hourly datasets, we employ the potential vorticity ($PV$), potential temperature ($\theta$), zonal and meridional wind speeds ($u$, $v$) to compute the PVG tropopause, as explained in Section 2.2. The PVG tropopause is compared to the WMO thermal definition (Section 2.3). Variability and trends of the tropopause are examined via multilinear regression, as detailed in Section 2.4. To reduce computational effort and improve usability, we explore simplifications of the method, which are explained in Section 2.5. We conclude with a regional analysis of the PVG tropopause climatology and trends, which is described in Section 2.6."

*Line 147: Why does the analysis only go through 2017?*

We chose the period 1980-2017 for this analysis as it is the maximum common period for which we have the data from all four reanalysis available.

*Line 175: I don't see the Bosilovich refence in the bibliography, but I think a more appropriate reference is Gelaro et al.*

I seem to have mixed up the references there – thanks for pointing that out! Fixed now.

*Line 180: What is the temporal resolution of the reanalysis data sets used here? The issues regarding the use of full resolution reanalysis data versus isentropic-interpolated monthly mean data could be at least partially addressed in this paragraph.*

Thank you! We changed the sentence to:

"The PVG tropopause is calculated based on the zonal and meridional wind speeds ($u$ and $v$), potential vorticity ($PV$) and potential temperature ($\theta$) from six-hourly reanalysis data. A computationally faster alternative using monthly climatologies of reanalyses is discussed in Section 2.5 below."

*Line 197: I think the lapse rate should be -dT/dz here?*

Yes, thank you! Changed that in the manuscript.

*Line 200 and 204: Would it be helpful to be more specific about the dimension (vertical or horizontal) which is being referred to here in the context of gradients?*

Thank you for pointing that out! Changed the wording to "meridional PV gradients" in both lines.

*Line 215: One thing that I was curious about in the PVG tropopause definition here is why both windspeed and PVG are used rather than just simply PVG. Perhaps this is addressed in the Kunz paper but could the authors mention somewhere in the paper why both are necessary? This relates to my comments around whether or not one really needs to implement the most maximalist analysis of the data in order to capture the important variability, or whether a more simplified definition (e.g., based on monthly mean PV data alone) would suffice?*

Thank you for this insightful remark! In Nash et al. (1996), where the idea of using the PV gradient and wind speed to diagnose the edge of the polar vortex was first introduced, the wind speed is included because multiple prominent maxima can occur in both the PV gradient and wind speed, but only one common peak appears at the vortex edge. Therefore, including the wind speed reduces noise in the diagnostic. We tested this for the PVG tropopause by running the algorithm omitting the wind criterion, and indeed found that using only PV gradient leads to a substantial increase in noise compared to considering the product of PV gradient and wind – see the new Methods section 2.5 and Results 3.5.

*Line 228: Again, going to the simplicity of the definition, is it really necessary to compute the break on equivalent latitude rather than geographical latitude?*

This is also a good question! Considering geographical latitude instead of equivalent latitude is a substantial change to the method introduced by Nash et al (1996) and Kunz et al (2011). We tested the algorithm on zonal mean reanalysis data and computed the PV gradient with respect to geographical latitude, which yielded substantially different results than computing the PV gradient from globally resoved data and equivalent latitude (see Methods 2.5, Results 3.5). We therefore conclude that these two methods are not comparable, and computing equivalent latitude is necessary.

*Line 235 – 248: These seem like a complicated set of rules and bring up several questions to me. First, do the authors really need to identify the PVG tropopause width at each reanalysis time step? If the goal is to create monthly means, why not average fields first? I suspect that doing something like this could allow the peak finding to be more straightforward and not require so many ad hoc rules.*

Indeed, the rules appear complicated. Thank you for the suggestion! We tested using monthly mean data, which yielded similar results than using sub-daily data, see method chapter 2.5 and results 3.5. Considering the set of rules, these have shown to be still necessary when using monthly climatologies, because strong wind and PVG maxima also appear near the polar vortex, which requires a poleward latitudinal boundary and choosing the most equatorward maximum. In order to mitigate noise, we kept the relative prominence criterion of maxima.

*Also, related to peak finding, there are several methods outlined in the TropD software package (Adam et al., 2018). Is there a reason the authors chose a different approach rather than adopting one of the well documented methods outlined there?*

Thank you for suggesting this software package, this is a valuable resource. Line 27 in Adam et al (2018) states that the methods are designed for zonal means and is not tested yet on gobally resolved fields. While we are using globally resolved fields, this would require further testing with our method and is out of the scope of our current study. We consider using the method for maximum determination based on weighted latitudes (Eq. 1, Adams et al. 2018) in one of our next studies and compare with our current algorithm. Thanks again for the suggestion!

*Line 265-267: I find the discussion around the seasonal cycle term confusing here. It seems as though it is something that is fit from the fact that it is included in the equation, but then it sounds like it is not included in the fit and that what is fit to is actually the de-seasonalized data.*

Thank you! The seasonal cycle $S(t)$ is indeed computed in advance and used as a regressor. The multilinear regression takes as input the time series, as well as $S$, $QBO_{30}$, $QBO_{50}$ and $ENSO$ and outputs the coefficients $a_{...}$. We added a sentence in line 274 which hopefully clarifies this:

"The regressors $S$, $QBO_{30}$, $QBO_{50}$ and $ENSO$ are determined in advance of the multilinear fit:"

*Line 276: The QBO winds at 30 and 50 hPa are not truly orthogonal to one another, and I think that a better practice is to fit to QBO EOFs. That said, my guess it doesn't significantly impact the results that much.*

Indeed, fitting to QBO EOFs would be a good alternative approach. We chose 30 and 50 hPa winds as QBO

regressors, as these have been frequently used in similar studies
(e.g. Stiller et al., 2012, ACP, https://acp.copernicus.org/articles/12/3311/2012/).

*Figure 1: I appreciate that it is easier to look at 2 plots than 4, but I always feel like I am missing something when people only show 2 seasons rather than 4. Given that the authors are trying to document a new method, I think it is important to show results for all 4 seasons. This could easily be done with the use of supplemental figures if the authors feel like it would significantly detract from the manuscript.*

You are right! We originally chose to show only the solstice seasons, since the changes in climatologies, as well as the differences in reanalyses, are strongest between DJF and JJA. However, we acknowledge that showing the equinoctial seasons is important as well, and included additional plots for MAM and SON corresponding to figures 1–4 (A1, A2, A3 and A4) in the appendix.

*Line 322: It looks to me like PVG and WMO tropopauses agree well up to 360K, but not 370K in all seasons/hemispheres.*

Indeed, thank you, that was a typo! Changed that to 360 K.

*Line 376-378: Is this due to STJ/EDJ separation?*

Thanks for the idea – that is likely the case and we added a corresponding sentence:

"The larger PV variance in summer is likely due to the weakening of the subtropical jet and coalescence with the eddy-driven jet (Manney and Hegglin, 2018)."

*Line 426-435: Is this due to horizontal resolution differences? Line 460-461: Do you have an idea of why ERA5 is different. Horizontal resolution?*

These are interesting questions which we also tried to find answers to in the literature. However, we currently have no certainty as to why PV differs between the reanalyses. This would probably require further in-depth analyses, which are out of the scope of our current study.

*Line 439: The word "Conclusively" doesn't seem right here*

Thank you for pointing that out! Changed the sentence to:

"We therefore advise using the same reanalysis for comparisons between the PVG tropopause and other variables."

*Line 585: "Hereof" doesn't seem like the correct word to use.*

That indeed sounds a little awkward! Reworded the sentence to:

"The seasonal cycle accounts for most of the variability, shifting the tropopause north- and southward by $5°$ to $15°$ latitude, which is concurrent with the shift of the ITCZ."

---

## Author Response (AR1)

**Author's Response: PVG Tropopause (egusphere-2024-471)**

We thank the Reviewers for the careful reading and evaluation of the manuscript and the good comments which helped a lot to further improve the paper. In the following, we address all comments and questions raised (Reviewer's comments in italics) and state all changes in the manuscript. Line numbers refer to the *latexdiff* file.

We see two main concerns raised by the Reviewers, regarding (i) the complexity of the PVG tropopause determination algorithm and (ii) the analyses being carried out in zonal mean view only, which masks regional variations. A short overview of the related changes in the revised manuscript is:

**(i) Simplification of the algorithm**

We acknowledge the concern regarding the computational expense of determining the PVG tropopause from sub-daily global reanalysis data with several additional criteria concerning the height of maxima, latitude boundaries, and wind speed. In response, we have conducted additional analyses to examine the feasibility of simplified methods using monthly averaged data and the potential impact on the accuracy of the PVG tropopause latitude estimation. Specifically, we tested the following alterations:

1. Monthly reanalysis climatologies

2. Monthly reanalysis climatologies without the wind criterion

3. Monthly and zonal mean reanalysis climatologies

4. Monthly and zonal mean reanalysis climatologies without the wind criterion

These methods were applied to ERA5 reanalysis data, and we compared the results with the standard method that uses six-hourly, global reanalysis data with the wind criterion.
Our findings indicate that using monthly averages does not significantly alter the climatological latitude of the PVG tropopause between 340K and 370K, though deviations are observed at lower and higher isentropes. Omitting the wind criterion leads to increased noise, corroborating findings by Nash et al. (1996), due to the stabilizing redundancy of the PV gradient and horizontal wind peaks. Zonally averaging the reanalysis data and computing the PV gradient with respect to geographical latitude instead of equivalent latitude resulted in substantial changes to the results, as the method is designed to work with equivalent latitudes (cf. Nash et al. 1996, Kunz et al. 2011).
Based on these results, we do not recommend using zonal mean reanalyses or omitting the wind criterion. Averaging reanalyses over months yields similar results for the tropopause climatology near the tropopause break between 340K and 370K, but notable deviations above and below. The trends computed from monthly climatologies qualitatively show a similar vertical structure, but deviate numerically from the trends computed from sub-daily data. Therefore, using monthly climatologies may be sufficient if the focus is on the climatological structure of the tropopause break, but trends should be computed with sub-daily data.
The addidtional analyses of method simplifications are detailed in two new sections in the manuscript: Methods section 2.5 and Results section 3.5.

**(ii) Regional climatology and long-term trend analysis**

We agree that using a global zonal mean in our analysis could mask significant regional (longitudinal) differences. To address this, we have conducted a supplementary regional analysis focusing on these variations in the climatology and long-term trends of the PVG tropopause.
By identifying the PV contour for the tropopause in ERA5 data at each isentropic level, we determined global surfaces of the tropopause $\theta(\phi, \lambda)$. Poleward of the 320K contour, we continued the PVG tropopause by setting the tropopause as the PV isosurface corresponding to the PV value of the tropopause on 320K.
We computed zonally resolved long-term trends by applying multilinear regression to the tropopause intersection at each longitude and isentrope. The results confirm that the mid-latitude tropopause widens poleward overall in both hemispheres, while the tropical tropopause narrows equatorward. These trends vary noticeably with longitude. Comparison with Martin et al. (2020) reveals a similar longitudinal pattern; specifically, the strongest narrowing trends are observed over the east Pacific.
The corresponding methods and results are described in additional subsections of the Methods chapter 2.6 and Results 3.6.

A more detailed reply to all comments and description of the changes in the revised manuscript is given below.

**1 Reviewer 1**

**1.1 Overall comments**

**R1:** *Tropical circulation and so-called "tropical width" changes are important phenomena to better understand in the context of climate variability and change. The paper "Variability and trends in the PV-gradient dynamical tropopause" by Turhal et al. examines a new definition for tropical width based on a potential vorticity-based definition that is applied to several reanalyses. This paper is very well written and thorough, and I have no doubt it will be of great interest to ACP readers. I have mostly minor suggestions to clarify a few points in the manuscript.*

**Authors:** Thank you for this positive evaluation of the manuscript!

**1.2 General comments**

**R1:** *The only "major" point I would like to suggest is that the authors consider whether their rather more involved methodology (which involves computing PV, etc. from the reanalysis model level fields at high temporal frequency and complicated peak finding rules) is really necessary in order to accurately quantify the seasonal to multidecadal variability in the PVG tropopause latitudes? I have no objection to applying this more "expensive" approach to analyzing the data if it is truly necessary, but it would be helpful for future users to know whether it really is necessary, or whether more simplified approaches with averaged fields could be used without a loss of fidelity (e.g., monthly mean PV/wind fields on pre-determined isentropic levels, as is commonly provided by reanalysis centers)? The answer to this question has implications for how easily this type of analysis might be extended in time or to other reanalyses, as well as how straightforwardly it might be applied to model output fields from multi model experiments (where high temporal / vertical resolution fields are often not made available). If the authors could provide some insight on this issue, I think it would be a valuable contribution to the community.*

**Authors:** This is a very good point! Determining the PVG tropopause from sub-daily reanalysis data over 40 years is indeed computationally expensive. As explained above in our main point (i), we have therefore run additional tests with monthly climatologies of the ERA5 reanalysis and examined the influence of using zonal mean data and omitting the wind criterion, as detailed in a new section 2.5 in the Methods chapter. We computed the seasonal climatologies, mean seasonal cycle and long-term trends from these additional runs, which are presented in a new section "3.5 Simplifications of the PVG tropopause determination method" in the results chapter.

**1.3 Specific comments**

**R1:** *Line 53: I suggest defining PVU at its first use (i.e., the SI unit equivalent)*

**Authors:** Thank you!

**Changes:** Rephrased the paragraph at lines 44-48 as follows and included the unit definition of PVU:

"Another common definition of the tropopause, the so-called *dynamical tropopause*, is based on the *potential vorticity (PV)*, an analogue of angular momentum in air flow introduced by Rossby (1940) and Ertel (1942). PV is measured in *Potential Vorticity Units (PVU)* with $1\,\mathrm{PVU} = 10^{-6}\,\mathrm{m^2 s^{-1} K\,kg^{-1}}$. An invertibility principle holds which allows inferring the flow velocity field from the PV distribution. PV is therefore closely linked to atmospheric dynamics (Hoskins, 1985), which makes it particularly valuable for transport studies. "

**R1:** *Line 84: Suggest citing Waugh et al 2018 here.*

**Authors:** Thank you for the suggestion!

**Changes:** Lines 89/90: Replaced the other references with "Waugh et al, 2018".

**R1:** *Line 97: Suggest citing Santer et al 2003 here*

**Authors:** Added the reference, thank you!

**Changes:** Lines 104/105: Added reference "Santer et al. 2003"

**R1:** *Line 114: I think "contracted" might be a more appropriate word here than "converged"*

**Authors:** Thanks for pointing out this misleading wording!

**Changes:** Lines 124/125: Changed wording to "while the STJ of both hemispheres converged towards each other".

**R1:** *Line 119: "to which extent" → "to what extent"*

**Authors:** Fixed!

**Changes:** Line 129

**R1:** *Line 140 − 145: This intro section confused me at first because it seemed to not be a detailed enough description, but then I eventually figured out that the material was discussed in further detail in the following subsections. You might consider referencing the subsections here to make it clear that further details are provided there.*

**R1:** *Line 140 − 141 (and elsewhere): I found myself confused a few times in the manuscript regarding some of the uses of words like "gradient" where the direction of the gradient was not really specified. In this sentence (and throughout the paper) the gradients being referenced are meridional gradients on isentropes. Similarly, there's a bit of potential confusion around the discussion of the PVG dynamical tropopause being something that is defined based on a meridional gradient, versus what most people think of which is the tropopause being defined based on a vertical gradient/threshold. It might be worth considering the language in the paper and whether it would be more appropriate to use the term PVG dynamical tropopause \*width\* when referring to the latitude, rather than just calling it the PVG dynamical tropopause.*

**Authors:** Indeed! We reworked the methods section and added references to the individual subchapters, and clarified that we compute a meridional gradient.

**Changes:** Lines 150–159:
"The PV gradient-based (PVG) dynamical tropopause (Kunz, 2011) is determined as a contour on surfaces of equal potential temperature (i.e., *isentropes*) from the meridional gradient of potential vorticity (PV), combined with the location of the subtropical jet streams. This study is based on four different meteorological reanalyses: ERA-Interim, ERA5, MERRA-2 and JRA-55, which are described in Section 2.1. From six-hourly datasets, we employ the potential vorticity ($PV$), potential temperature ($\theta$), zonal and meridional wind speeds ($u$, $v$) to compute the PVG tropopause, as explained in Section 2.2. The PVG tropopause is compared to the WMO thermal definition (Section 2.3). Variability and trends of the tropopause are examined via multilinear regression, as detailed in Section 2.4. To reduce computational effort and improve usability, we explore simplifications of the method, which are explained in Section 2.5. We conclude with a regional analysis of the PVG tropopause climatology and trends, which is described in Section 2.6."

**R1:** *Line 147: Why does the analysis only go through 2017?*

**Authors:** We chose the period 1980-2017 for this analysis as it is the maximum common period for which we have the data from all four reanalysis available.

**R1:** *Line 175: I don't see the Bosilovich refence in the bibliography, but I think a more appropriate reference is Gelaro et al.*

**Authors:** I seem to have mixed up the references there – thanks for pointing that out! Fixed now.

**Changes:** Replaced reference in line 189.

**R1:** *Line 180: What is the temporal resolution of the reanalysis data sets used here? The issues regarding the use of full resolution reanalysis data versus isentropic-interpolated monthly mean data could be at least partially addressed in this paragraph.*

**Authors:** Thank you!

**Changes:** Lines 201–202: Changed the sentence to:

"The PVG tropopause is calculated based on the zonal and meridional wind speeds ($u$ and $v$), potential vorticity ($PV$) and potential temperature ($\theta$) from six-hourly reanalysis data. A computationally faster alternative using monthly climatologies of reanalyses is discussed in Section 2.5 below."

**R1:** *Line 197: I think the lapse rate should be -dT/dz here?*

**Authors:** Yes, thank you! Changed that in the manuscript.

**Changes:** Line 214: Added minus sign

**R1:** *Line 200 and 204: Would it be helpful to be more specific about the dimension (vertical or horizontal) which is being referred to here in the context of gradients?*

**Authors:** Thank you for pointing that out!

**Changes:** Changed the wording to "meridional PV gradients" in lines 217 and 222.

**R1:** *Line 215: One thing that I was curious about in the PVG tropopause definition here is why both windspeed and PVG are used rather than just simply PVG. Perhaps this is addressed in the Kunz paper but could the authors mention somewhere in the paper why both are necessary? This relates to my comments around whether or not one really needs to implement the most maximalist analysis of the data in order to capture the important variability, or whether a more simplified definition (e.g., based on monthly mean PV data alone) would suffice?*

**Authors:** Thank you for this insightful remark! In Nash et al. (1996), where the idea of using the PV gradient and wind speed to diagnose the edge of the polar vortex was first introduced, the wind speed is included because multiple prominent maxima can occur in both the PV gradient and wind speed, but only one common peak appears at the vortex edge. Therefore, including the wind speed reduces noise in the diagnostic. We tested this for the PVG tropopause by running the algorithm omitting the wind criterion, and indeed found that using only PV gradient leads to a substantial increase in noise compared to considering the product of PV gradient and wind – see the new Methods section 2.5 and Results 3.5.

**R1:** *Line 228: Again, going to the simplicity of the definition, is it really necessary to compute the break on equivalent latitude rather than geographical latitude?*

**Authors:** This is also a good question! Considering geographical latitude instead of equivalent latitude is a substantial change to the method introduced by Nash et al (1996) and Kunz et al (2011). We tested the algorithm on zonal mean reanalysis data and computed the PV gradient with respect to geographical latitude, which yielded substantially different results than computing the PV gradient from globally resoved data and equivalent latitude (see Methods 2.5, Results 3.5). We therefore conclude that these two methods are not comparable, and computing equivalent latitude is necessary.

**R1:** *Line 235 – 248: These seem like a complicated set of rules and bring up several questions to me. First, do the authors really need to identify the PVG tropopause width at each reanalysis time step? If the goal is to create monthly means, why not average fields first? I suspect that doing something like this could allow the peak finding to be more straightforward and not require so many ad hoc rules.*

**Authors:** Indeed, the rules appear complicated. Thank you for the suggestion! We tested using monthly mean data, which yielded similar results than using sub-daily data, see method chapter 2.5 and results 3.5. Considering the set of rules, these have shown to be still necessary when using monthly climatologies, because strong wind and PVG maxima also appear near the polar vortex, which requires a poleward latitudinal boundary and choosing the most equatorward maximum. In order to mitigate noise, we kept the relative prominence criterion of maxima.

**R1:** *Also, related to peak finding, there are several methods outlined in the TropD software package (Adam et al., 2018). Is there a reason the authors chose a different approach rather than adopting one of the well documented methods outlined there?*

**Authors:** Thank you for suggesting this software package, this is a valuable resource. Line 27 in Adam et al (2018)

states that the methods are designed for zonal means and is not tested yet on gobally resolved fields. While we are using globally resolved fields, this would require further testing with our method and is out of the scope of our current study. We consider using the method for maximum determination based on weighted latitudes (Eq. 1, Adams et al. 2018) in one of our next studies and compare with our current algorithm. Thanks again for the suggestion!

**R1:** *Line 265-267: I find the discussion around the seasonal cycle term confusing here. It seems as though it is something that is fit from the fact that it is included in the equation, but then it sounds like it is not included in the fit and that what is fit to is actually the de-seasonalized data.*

**Authors:** Thank you! The seasonal cycle $S(t)$ is indeed computed in advance and used as a regressor. The multlinear regression takes as input the time series, as well as $S$, $QBO_{30}$, $QBO_{50}$ and $ENSO$ and outputs the coefficients $a_{...}$.

**Changes:** We added a sentence in lines 286/287 which hopefully clarifies this:

"The regressors $S$, $QBO_{30}$, $QBO_{50}$ and $ENSO$ are determined in advance of the multilinear fit:"

**R1:** *Line 276: The QBO winds at 30 and 50 hPa are not truly orthogonal to one another, and I think that a better practice is to fit to QBO EOFs. That said, my guess it doesn't significantly impact the results that much.*

**Authors:** Indeed, fitting to QBO EOFs would be a good alternative approach. We chose 30 and 50 hPa winds as QBO regressors, as these have been frequently used in similar studies
(e.g. Stiller et al., 2012, ACP, https://acp.copernicus.org/articles/12/3311/2012/).

**R1:** *Figure 1: I appreciate that it is easier to look at 2 plots than 4, but I always feel like I am missing something when people only show 2 seasons rather than 4. Given that the authors are trying to document a new method, I think it is important to show results for all 4 seasons. This could easily be done with the use of supplemental figures if the authors feel like it would significantly detract from the manuscript.*

**Authors:** You are right! We originally chose to show only the solstice seasons, since the changes in climatologies, as well as the differences in reanalyses, are strongest between DJF and JJA. However, we acknowledge that showing the equinoctial seasons is important as well, and included additional plots for MAM and SON corresponding to figures 1–4 (A1, A2, A3 and A4) in the appendix.

**R1:** *Line 322: It looks to me like PVG and WMO tropopauses agree well up to 360K, but not 370K in all seasons/hemispheres.*

**Authors:** Indeed, thank you, that was a typo!

**Changes:** Lines 362 and 383: Changed to 360 K.

**R1:** *Line 376-378: Is this due to STJ/EDJ separation?*

**Authors:** Thanks for the idea – that is likely the case.

**Changes:** Lines 440/441: Added a corresponding sentence:

"The larger PV variance in summer is likely due to the weakening of the subtropical jet and coalescence with the eddy-driven jet (Manney and Hegglin, 2018)."

**R1:** *Line 426-435: Is this due to horizontal resolution differences? Line 460-461: Do you have an idea of why ERA5 is different. Horizontal resolution?*

**Authors:** These are interesting questions which we also tried to find answers to in the literature. However, we currently have no certainty as to why PV differs between the reanalyses. This would probably require further in-depth analyses, which are out of the scope of our current study.

**R1:** *Line 439: The word "Conclusively" doesn't seem right here*

**Authors:** Thank you for pointing that out!

**Changes:** Line 500: Changed the sentence to:

"We therefore advise using the same reanalysis for comparisons between the PVG tropopause and other variables."

**R1:** *Line 585: "Hereof" doesn't seem like the correct word to use.*

**Authors:** That indeed sounds a little awkward!

**Changes:** Line 727: Reworded the sentence to:

"The seasonal cycle accounts for most of the variability, shifting the tropopause north- and southward by $5°$ to $15°$ latitude, which is concurrent with the shift of the ITCZ."

**2 Reviewer 2**

**2.1 Overall comments**

**R2:** *This paper uses a PV-gradient tropopause definition to assess the climatology, variability, and long term trends of the tropopause in potential temperature space, with applicability to assessing tropical width. The authors' method is novel in allowing for vertical variation of tropical width and their results fit in well with the current state of the research. The paper is thorough and compelling and will be of substantial interest to the readers of ACP. There are some places where I feel the manuscript could be more clear and concise to improve the readability of the paper, for which I have provided some suggestions below. Additionally, I find the global-zonal mean aspect of the analysis to be a limitation of the work which should either be addressed by additional analysis or further discussion. My detailed comments are below.*

**Authors:** Thank you for this positive evaluation of the manuscript and your constructive suggestions! Our responses to your comments are detailed below.

**2.2 General comments**

**R2:** *The use of a global zonal mean in all the analysis seems limiting for the work, especially given some of the citations provided that emphasized that there were strong regional (longitudinal) differences in assessing STJ/tropical width changes. Specifically, the asian monsoon anticyclone has such a profound impact on the tropopause height and the lower stratosphere in general in the northern hemisphere that is lost in this analysis because of global zonal means. Additionally, some of the tropopause characteristics of the northern hemisphere might be skewed because of the influence of the asian monsoon. I think some supplementary regional analysis would greatly improve the paper. Otherwise, there should be some discussion about these limitations.*

**Authors:** Thank you for this valuable idea! We agree that using a global zonal mean in our analysis could mask significant regional (longitudinal) differences as observed by Manney and Hegglin (2018) and Martin et al. (2020). As described above in our general point (ii), we therefore incorporated a regional analysis, which is detailed in a new section 2.6 "Regional aspects" of the Methods chapter; results are presented in the new Results section 3.6.

**R2:** *Current figure captions should probably all include that they are zonal means for clarity*

**Authors/Changes:** Added the phrase "averaged over equivalent latitude contours" to Figs. 1–11.

**R2:** *I suggest eliminating the uses of "cf." throughout the manuscript in references to figures/equations, as this suggests that you are wanting readers to "compare" with your own works. In many of these situations just deleting the cf. and using (Fig. X) or (Eq. Y) is appropriate. In other cases, I suggest replacing (cf. Fig. X) with (see Fig. X)*

**Authors:** Thanks for the suggestion, I wasn't aware of that.

**Changes:** Deleted/ replaced "cf." throughout the manuscript as suggested.

**R2:** *There are several uses of the word "significant" throughout the manuscript that may be more suitably expressed as "substantial" or a similar word, since "significant" implies the existence of statistical significance testing.*

**Authors:** Thank you for pointing that out!

**Changes:** Replaced "significant" with "substantial" throughout the manuscript.

**R2:** *There are several uses of the word "resolution" throughout the manuscript that I believe is intended to describe horizontal or vertical grid spacing and therefore should be replaced with "grid spacing"*

**Authors:** Indeed, thanks!

**Changes:** Replaced "resolution" with "grid spacing" throughout the manuscript.

**R2:** *What time steps are used in your analysis? Are they all at the reanalysis native time steps (6 hourly except for ERA 5 at 1 hourly)? Or are they all 6 hourly? Or daily? If you are working at every timestep, does the increased temporal frequency make a difference?*

**Authors:** We used every reanalysis in six-hourly time steps – sorry for not clarifying that earlier!

**Changes:** Added this to the sentence starting at line 201:

"The PVG tropopause is calculated based on the zonal and meridional wind speeds ($u$ and $v$), potential vorticity ($PV$) and potential temperature ($\theta$) from six-hourly reanalysis data."

**R2:** *With the value of PV being important and frequently discussed throughout the paper, it makes me wonder if the strength of the gradient is important? And how would that vary seasonally and over time? This may be outside of the scope of this work but seems just as important as PV magnitude when thinking about relevance to dynamic transport.*

**Authors:** This is a very interesting suggestion! In our following research, we plan to examine tracer concentrations and trajectories in relation to the PVG tropopause using a Lagrangian model, where including the magnitude of PV gradient would certainly grant additional insights.

**R2:** *I found some of the differences between reanalyses surprising. Some discussion into possible reasonings could be helpful. Especially when ERA-5 is quite a bit different from the others... is it because of resolution?*

**Authors:** This is indeed a surprising result. A well-grounded answer would require further in-depth analyses, which are out of the scope of our current study. Here we can only speculate that these differences are likely related to the significantly higher resolution of ERA5 compared to the other reanalyses.

**2.3   Specific comments**

**R2:** *Lines 1-2: This opening sentence was a little confusing to read at first. I suggest rewording to "The dynamical tropopause acts as a transport... lowermost stratosphere and is characterized by..." or something similar*

**Authors:** Your phrasing is indeed much more readable.

**Changes:** Lines 1–2: Reworded the sentence to:

"The dynamical tropopause acts as a transport barrier between the tropical upper troposphere and extratropical lowermost stratosphere and is characterized by steep gradients in potential vorticity (PV) along an isentropic surface."

**R2:** *Line 34: suggest adding "at altitudes" before "slightly below the subtropical jets"*

**Authors:** Done!

**Changes:** Line 36

**R2:** *Lines 35-36: Do these citations present estimated frequencies of these occurrences that could be included here?*

There are no mentions of the frequency of common EDJ/STJ maxima in the literature cited here; this was only mentioned in the text and apparent from figures.

**R2:** *Line 38-41: I wouldn't say the lowermost stratosphere plays a role in STE as much as it is an important location with regards to STE. Suggest rewording.*

**Authors:** Thank you for the suggestion!

**Changes:** Rephrased line 41:

"which is an important region with regards to stratosphere-troposphere exchange (STE)"

**R2:** *Line 46: Suggest rewording to "At the tropopause, there is a transition from lower PV values..."*

**Authors:** Thanks!

**Changes:** Rephrased line 50 as suggested.

**R2:** *Line 55: 'Applicability' might be a better word than 'accuracy' here?*

**Authors:** Indeed!

**Changes:** Line 60: Replaced "accuracy" with "applicability".

**R2:** *Lines 87-91: This sentence is a bit long and hard to understand. Specifically, I'm not sure exactly if the statement "and coupling with sea surface temperatures" is supposed to be related to the Hadley cell or jet locations or ENSO or the PDO or all of the above.*

**Authors:** Thank you!

**Changes:** Broke up the sentence starting at line 93 into multiple sentences for clarity.

**R2:** *Line 91: Suggest rewording "It is even under discussion"*

**Authors:** Thank you!

**Changes:** Reworded the sentence starting at line 100 to:

"The role of internal variability is still under discussion; some studies indicate that the impact of natural variability even exceeds that of long-term changes, and that the currently available 40 years of reanalysis data are too short to discern forced expansion from natural variability [citations]."

**R2:** *Paragraph starting at Line 94: Discussion of double tropopauses and their relationship to a changing climate/tropical width could be a nice addition here*

**Authors:** Thank you for the suggestion! However, we chose to not include double tropopauses in out discussion, since we do not consider these in our analysis.

**R2:** *Section 2.1: The approximate vertical resolution of each reanalysis in the region of interest (UTLS) would be a helpful addition here*

**Authors:** Thank you!

**Changes:** Added a new sentence to line 194:

"The vertical spacing of the native model grids around the tropopause is approximately $1\,\mathrm{km}$ for ERA-Interim, JRA-55 and MERRA-2 and $0.3\,\mathrm{km}$ for ERA5, transformed from pressure to log-pressure altitude (Fujiwara et al. 2022, cf. Fig. 2.1)."

**R2:** *Line 175: I think Gelaro et al. 2017 is the appropriate reference here (https://doi.org/10.1175/JCLI-D-16-0758.1)*

**Authors:** Thank you, I mixed up the references!

**Changes:** Line 189: Replaced with Gelaro et al.

**R2:** *Line 183: I don't think "down-scaled" is the right word here. Also, is the vertical resolution changed at all in this run?*

**Authors/Changes:** Line 198: Removed "down-scaled" and reworded the sentence to

"ERA5 data is used with a $1° \times 1°$ latitude-longitude grid as provided by the ECMWF"

**R2:** *Section 2.2: This section is where I felt the manuscript most lacked clarity and conciseness. I understand and sympathize with the fact that there are many important details that should be included here, but there were some places where it felt some of these details overshadowed the main goal of what you were trying to show and emphasize here. One potential suggestion would be to carefully consider which equations actually need to be shown here and described in detail. And/or, I think it would be possible to shorten some of the discussion on page 7 without taking away from the important points you are wanting to get across.*

**Authors:** Thank you for your suggestions! We acknowledge that the explanations on the PVG tropopause are lengthy and contain many equations. However, since the PVG tropopause is not yet well-established, we feel that a detailed, comprehensive explanation of the theoretical framework explaining the relationship between PV, the STJ, and the transport barrier is necessary to understand the origin of the PVG tropopause concept. Therefore, we have chosen not to alter this section.

**R2:** *Some of the details in this section also confused me a bit about the inclusion of the importance of wind shear in calculating PV. Its emphasized in line 203-207 that relative vorticity is strongly impacted by horizontal wind shear and therefore the STJs. This led me to question whether the inclusion of horizontal wind in calculating Q sort of 'double counts' the STJ in a way? I think clearer discussion on this would be helpful.*

**Authors/Changes:** This is a good observation! We mentioned the redundancy of including the subtropical jet and PV gradient in the Results chapter 3.5 on simplifications of the method (line 655):

"In addition, the STJ wind criterion in Eq. 5 is necessary, because the redundancy of the horizontal wind and PV gradient maxima at the tropopause break stabilizes the PVG tropopause calculation."

**R2:** *Line 197: lapse rate should be negative dT/dZ*

**Authors:** Thank you, added a minus sign there.

**Changes:** Line 214

**R2:** *Figure 1: Specify that this is a global zonal mean.*

**Authors:** Thanks, done.

**Changes:** Added "averaged over equivalent latitude contours" to the description of Figs. 1, 3 and 5–9. (Figures 3–9 are not shown in the latexdiff document for unknown reasons; we were not able to fix that.)

**R2:** *Lines 298 - 300: This sentence is a bit wordy, suggest revising to something along the lines of "Hence, the common practice of defining the dynamical tropopause as a PV isosurface (e.g., 2 PVU) for studies of stratosphere-troposphere exchange..."*

**Authors:** Thank you, reworded according to your suggestion.

**Changes:** Reworded sentence starting in line 359.

**R2:** *Lines 300-301: I would be careful saying that they agree well throughout different seasons when MAM and SON are both not shown here*

**Authors:** Thank you for the suggestion! Added plots for MAM and SON.

**Changes:** Added figures showing the equinoxes corresponding to Figs. 1–4 and Fig. 13 in the appendix (Figs. A1–A4 and C1).

**R2:** *Line 350: Please provide reasoning for only including DJF and JJA. Is it just that the results from other seasons just not that interesting?*

**Authors:** The idea behind showing only DJF and JJA was to contrast the solstice seasons with each other, since the changes in climatologies, as well as the differences in reanalyses, are strongest between winter and summer. Additionally, we did not want to not overcrowd the manuscript with figures. However, we acknowledge that showing the equinoctial seasons is important as well, which is why we included additional plots for MAM and SON in the appendix.

**R2:** *Figure 2: The organizing of the panels on this figure (going a,b,e,f across the top row and c,d,g,h across the bottom row) feels a bit weird*

**Authors:** We understand that, however, we wanted to specifically group the subfigures into a–d for DJF and e–h for JJA.

**R2:** *Line 387: Please include a citation here*

**Authors:** "This seasonal offset towards positive PV likely corresponds to the northward shift of the ITCZ in June–August."

The reasoning behind this statement is that PV increases towards the poles due to the increase of the Coriolis parameter, and that a northward shift therefore should lead to an increase of PV. We did not find any literature specifically supporting this.

**R2:** *Line 417: I think "consistent" might be more appropriate than "robust" here*

**Authors:** Thanks! Reworded to "consistent".

**Changes:** Line 480

**R2:** *Line 420: Suggest rewording "could probably be related to"*

**Authors:** Thanks!

**Changes:** Line 486: Reworded to "are potentially related to".

**R2:** *Line 432: Maybe go with "qualitatively similar" or "qualitatively consistent" instead of robust*

**Authors/Changes:** Line 495: Changed to "qualitatively similar".

**R2:** *Line 439: Conclusively doesn't feel like the right word here.*

**Authors:** Thank you for pointing that out!

**Changes:** Reworded the sentence starting at line 503 to:

"Given the differences in PV values among the four reanalyses, we advise using the same reanalysis for comparisons between the PVG tropopause and other variables."

**R2:** *Lines 439-442: While I agree with the sentiment of this statement about avoiding errors by using one consistent reanalysis, I think it's also important to emphasize that use of multiple models could be important to represent the range of possibilities of the actual atmosphere due to some of the larger inconsistencies between the reanalyses.*

**Authors:** Thanks for this insightful comment! The intention of this sentence was to state that given the differences between PV values in different reanalyses, the PVG tropopause and other variables that one wants to compare to the tropopause should be consistently calculated from one reanalysis dataset. However, we fully agree with the Reviewer that considering a suite of models or reanalyses is important to represent the range of uncertainty in the state of the atmosphere. We changed the paragraph accordingly:

**Changes:** Starting in line 503: "Given the differences in PV values among the four reanalyses, we advise using the same reanalysis for comparisons between the PVG tropopause and other variables. In particular, the tropopause PV values need to be consistently calculated from one dataset. This mitigates the risk of introducing errors stemming from variations in the PV fields across different reanalyses. Additionally, comparing multiple reanalyses or models is important to represent the range of uncertainty in the state of the atmosphere."

**R2:** *Line 474: "PV variability appears smaller than expected..." compared to what? What was expected and why? Lines 474-478: suggest rewording this for clarity, maybe by breaking into two separate sentences?*

**Authors:** Thanks for pointing that out! This is indeed confusing.

**Changes:** Reworded the sentence starting in line 539:

"PV variability is small compared to the latitudinal variability between $320\,\mathrm{K}$ to $350\,\mathrm{K}$."

**R2:** *Figure 5: It could be useful here to plot the absolute value of pv on the ordinate to make it easier for the reader to see the direct comparison of the NH and SH? But this may just be a personal preference*

**Authors:** Thanks for the suggestion! We see the advantages of plotting the absolute PV, however in this case, we prefer to plot signed PV in order to visualize the north- and southward shifts of the tropopause in both hemispheres.

**R2:** *Figures 6,7,8: I suggest changing the order in which the reanalysis markers are plotted, such that the ERA-Interim "×" and the MERRA-2 "+" are plotted last and are therefore not covered up by the diamonds/circles.*

**Authors/Changes:** This is a good idea, thank you! Reworked the plots accordingly.

**R2:** *Figure 7: What does the "d." in "d. El Nino" mean in the panels on the top row? I'm not sure if this is commonplace and I just haven't seen it before, or if other readers would maybe be confused by this as well.*

**Authors:** Thank you for pointing that out, I was not aware of that! The "d" is an abbreviation for "during". Clarified this in the figure description:

**Changes:** Caption of Fig. 7: "The annotations *poleward/equatorward d. El Niño* aim to indicate the direction in which the tropopause is shifted during El Niño."

**R2:** *Line 491: Suggest changing "vertical structure in the different reanalyses" to "vertical structure consistent between the different reanalyses"*

**Authors/Changes:** Thanks! Reworded according to your suggestion (line 561).

**R2:** *Line 512: Do you include amplitudes of 0 - 1.5 degrees just because of the ERA-Interim outlier at the 320 K? I would maybe suggest changing this to say that amplitudes are primarily confined to 0-1 degree (or even smaller), since that is what is shown basically everywhere else. Do you have any thoughts as to why ERA-Interim is such an outlier at this level in both Figs. 7 and 8?*

**Authors:** Thanks for the suggestion! Since 1.5 degrees is indeed an outlier, we rephrased the sentence to "(...) with amplitudes primarily confined between $0°$ to $1°$ latitude". Concerning the reasons for this outlier, we can only speculate. We know that the 320K isentrope sometimes cuts into the orography – especially in summer –, causing

the method to fail at the corresponding times. However, we cannot pinpoint the reasons why this happens with some reanalyses more than with the other.

**R2:** *Lines 515-518: This is a really interesting result and is surprising to me. Is this something you expected or is consistent with previous work? Some discussion of this here or in the conclusion could be nice*

**Authors:** This was also a very surprising result for us. We have tried to find literature on dynamical links between the QBO and tropopause break/jet shifts, but could not find anything to support further discussion – therefore, we are careful to speculate on this result.

**R2:** *Lines 554-555: This concluding sentence feels a bit strong considering the large differences between the reanalyses.*

**Authors/Changes:** You are right. Replaced "robust" with "qualitatively consistent" in line 625.

**R2:** *Line 578: This first sentence feels repetitive with the following sentence. Suggest deleting or reworking*

**Authors/Changes:** Indeed! Deleted the sentence and rephrased the following sentence (starting in line 709) to:

"Examining the global isentropic PV fields (Fig. 4) in all reanalyses, we find that the latitude of the maximum PV gradient is consistent in all reanalyses, but PV values at the latitude of the maximum PV gradient vary."

**R2:** *Line 583: I think "largely robust variability" is a bit strong here*

**Authors/Changes:** Line 724: Rephrased to: "(...) reveals that the modes of variability are consistent in the four considered reanalysis, including (...)"

**R2:** *Line 585: suggest deleting or changing "hereof"*

**Authors/Changes:** Reworded the sentence starting in line 727 to:

"The seasonal cycle accounts for most of the variability, shifting the tropopause north- and southward by $5°$ to $15°$ latitude, which is concurrent with the shift of the ITCZ."

**R2:** *Lines 607-608: This seems very important and potentially deserving of a few more sentences*

**Authors/Changes:** This is mentioned in the introduction, lines 100-102.

**2.4 Technical corrections**

Many thanks for pointing out these mistakes!

**R2:** *Line 8: inter-annual → interannual*

**Authors/Changes:** Done (line 8).

**R2:** *Line 29: concurs → coincides*

**Authors/Changes:** Done (line 31).

**R2:** *Line 51: The PV-based → A PV-based*

**Authors/Changes:** Done (line 55).

**R2:** *Line 53: Define/include metric equivalent of PVU*

**Authors/Changes:** Thank you! Rephrased the paragraph at lines 44-49 as follows and included the unit definition of PVU:

"Another common definition of the tropopause, the so-called *dynamical tropopause*, is based on the *potential vorticity (PV)*, an analogue of angular momentum in air flow introduced by Rossby (1940) and Ertel (1942). PV is measured in *Potential Vorticity Units (PVU)* with $1\,\mathrm{PVU} = 10^{-6}\,\mathrm{m^2 s^{-1} K\,kg^{-1}}$. An invertibility principle holds which allows inferring the flow velocity field from the PV distribution. PV is therefore closely linked to atmospheric dynamics (Hoskins, 1985), which makes it particularly valuable for transport studies. "

**R2:** *Line 61: Don't capitalize upper troposphere lower stratosphere*

**Authors/Changes:** Fixed (line 62)!

**R2:** *Line 63: Furthermore → Hereafter*

**Authors/Changes:** Replaced accordingly (line 68).

**R2:** *Line 68: metrics are → metrics have been*

**Authors/Changes:** Corrected (line 73).

**R2:** *Line 125: As of now → To the authors knowledge*

**Authors/Changes:** Replaced accordingly (line 135).

**R2:** *Line 133: Remove dash in "time-scales"*

**Authors/Changes:** Done (line 143).

**R2:** *Line 156: ERA5 is extended from → ERA5 extends from*

**Authors/Changes:** Done (line 170).

**R2:** *Line 190: levels of potential temperature → surfaces of constant potential temperature*

**Authors/Changes:** Done (line 207).

**R2:** *Line 250: World Meteorological Organization WMO (1957) → World Meteorological Organization (WMO; World Meteorological Organization 1957)*

**Authors/Changes:** Done (line 269).

**R2:** *Line 275: normed → normalized*

**Authors/Changes:** Done (line 294).

**R2:** *Line 301: remove "ca."*

**Authors/Changes:** Changed to "up to 360 K" (line 362).

**R2:** *Line 353 and nearby: resolution → bin size*

**Authors/Changes:** Done (line 414).

**R2:** *Line 362: span → reach*

**Authors/Changes:** Done (line 423).

**R2:** *Line 483: Inter-annual → Interannual*

**Authors/Changes:** Done (line 553).

**R2:** *Line 498: similarly → similar*

**Authors/Changes:** Done (line 569).

**R2:** *Line 512: magnitudes → magnitude*

**Authors/Changes:** Done. Page not shown in latexdiff for unknown reasons, please refer to the reviewed pdf, line 566.

**R2:** *Line 567: fix the single quotation mark direction*

**Authors/Changes:** Line 697: Replaced with double quotation marks in the correct directions.

---

## Author Response (AR2)

**Author's Response to Editor Decision: PVG Tropopause (egusphere-2024-471)**

We thank the Editor for the careful reading of the manuscript and the helpful comments! In the following, we address all comments and questions raised (Editor's comments in italics) and state all changes in the manuscript. Line numbers refer to the *latexdiff* file.

**1 Overall comment**

**Editor:** *The authors have made a great effort to address the reviewers' comments on their manuscript, and have added a good deal of new analysis, which demonstrates that simplifed versions of their methodology may be appropriate for some uses. I have just a few remaining minor comments.*

**Authors:** Thank you for this positive evaluation of the manuscript!

**2 Specific comments and questions**

**Editor:** *Both reviewers asked why ERA5 was so different to the other reanalyses. You have speculated that this could be a reflection of the much higher resolution in ERA5. I understand that you don't want to speculate in the text. However, I wonder if your methodology contributes to this. All reanalyses are interpolated onto isentropic levels, with 10K spacing, except for ERA5, which is interpolated onto 5K levels. Do you still see marked differences between ERA5 and the other reanalyses if you also interpolate ERA5 onto 10K levels?*

**Authors:** Thank you for the interesting question! We discussed whether interpolation of ERA5 data from native model levels onto isentropic levels with a coarser spacing of 10 K instead of 5 K would mitigate the differences in PV observed between ERA5 and the other reanalyses. We came to the conclusion that this interpolation would not affect the PV values on each isentropic level and does therefore not explain the PV differences between reanalyses. We assume that the steeper PV gradient in ERA5 is an internal property of the reanalysis and maybe due to the higher native grid resolution, but we can not pinpoint this to any specific cause yet.

**Editor:** *Reviewer 1 asked why you chose 1980-2017 as your analysis period. I recommend adding your reasoning (that this is the period with the largest overlap between reanalysis datasets) to Section 2.1.*

**Authors:** Thank you for the suggestion! We reworked lines 153–154 and added the following sentence at the end of Section 2.1, lines 193–194:
"We consider the period from 1980 to 2017, as it provides the largest overlap of available post-processed reanalysis datasets at the time of this study."

**Editor:** *Reviewer 2 asked you to remove a 'ca.', which you changed to 'up to 360K'. Just checking that you didn't mean to change this to "~360K", which would be closer to your original meaning.*

**Authors:** Thanks for the idea! We changed this to "up to ~360K" in line 353.

**Editor:** *L283, please add here that the choice of 30 and 50hPa to capture the QBO follows Stiller et al., 2012*

**Authors:** Thanks! Added "following Stiller et al., 2012" to line 274.

**Editor:** *L506: Additionally → However*

**Authors:** Thanks, done, line 495.

**Editor:** *L540: There appears to be only half a sentence here. Please check.*

**Authors:** Thank you for the note! This is most likely due to the LaTeX diff file being broken. However, the sentence is complete in the submitted manuscript, see also line 528 in the new latexdiff file.

**Editor:** *L671: 'due to the wavy pattern of PV contours' isn't really an explanation of this behaviour. Can you link it better to other features of the atmospheric circulation, such as the jets, or the monsoon high?*

**Authors:** Indeed, thank you for the suggestion! We elaborated on the zonal variations of the tropopause as follows, lines 654–657:

"These global fields reveal noticeable zonal variations in tropopause height and latitude, which can be attributed to undulations of the subtropical jet streams caused by Rossby waves, as well as baroclinic instability forming high- and low-pressure areas, leading to zonal variability in the PV field."

**Editor:** *L734: During → during*

**Authors:** Thank you! Fixed, line 717.

**Editor:** *L785: three reviewers → two reviewers*

**Authors:** Thanks, reworked the sentence accordingly in lines 767–769:

"Last but not least, we greatly appreciate the careful reading of the manuscript and helpful feedback by the editor Laura Wilcox, two anonymous reviewers, as well as Tim Blazytko, Jan Conrads and Y. C.."